# See More, Forecast Better and Faster: Enhancing Time Series Foundation Models via Inference-Time Plug-and-Play Downsampling

Longlong Xu [1]  Zeyan Li [2]  Xiao He [2]  Zhaoyang Yu [2]  Dazhong Wen [1]  Mingze Sun [1]  Changhua Pei [3]  Dan Pei [1]

## Abstract

Time series foundation models (TSFMs) have demonstrated impressive generalization capabilities across diverse domains. However, they face significant challenges in long-term and ultra long-term forecasting. These challenges primarily arise from scalability limitations when TSFMs process extensive sequence lengths. To address this, we propose SPRINT, a training-free plug-and-play framework designed to empower TSFMs to see more, forecast better and faster during inference. The core idea is to perform forecasting in a downsampled-resolution space, enabling an extended look-back window with reduced computational costs. To avoid information loss and resolution mismatch caused by downsampling, SPRINT decomposes time series into trend and seasonal components, processing them separately. It predicts the low-frequency trend via a Resolution Interpolation workflow within the downsampled space, while preserving high-frequency details through a Pattern Replication mechanism for seasonality. Extensive experiments show that SPRINT achieves a significant improvement, increasing accuracy by 19% while enhancing efficiency with a reduction of max memory usage by $6.4\times$ and inference time by $16.9\times$ compared to state-of-the-art TSFMs. Code is available at https://github.com/NetManAIOps/SPRINT.

## 1. Introduction

Time series forecasting (Wen et al., 2022; Qiu et al., 2025) has been studied across widespread domains such as environment, economics, energy and meteorology. While deep learning models have achieved impressive performance, they typically require training from scratch on specific datasets, limiting their generalization across domains. Inspired by development in large models (Bommasani, 2021), pretrained time series foundation models (TSFMs) have recently emerged, demonstrating promising generalization capabilities across diverse tasks and domains.

Despite these advancements, real-world time series data across many domains is often sampled at high frequency. For instance, the recent MarmAudio dataset (Lamothe et al., 2025) consists of audio recordings sampled at 96 kHz. Even for minute-level data, a single daily cycle comprises 1,440 points, making the capture of weekly or monthly patterns require processing tens of thousands of data points. However, existing TSFMs face significant challenges when handling such long sequences. In the research community (Wu et al., 2023; Wang et al., 2024), forecasting involving extensive prediction lengths are typically categorized as *long-term* or even *ultra-long-term* forecasting. As look-back and prediction lengths increase, TSFMs encounter an efficiency bottleneck due to their inherent complexity. Additionally, zero-shot TSFMs often struggle with accurate long-term forecasting due to the volatility and complex long-range dependencies inherent in high-resolution sequences.

Crucially, time series data often exhibits significant redundancy. As evidenced by frequency-domain compression and filtering strategies, complex signals can often be accurately reconstructed from a reduced set of critical components (Shannon, 2006; Oppenheim, 1999). It suggests that core information in time series can be preserved even with reduced data density. Based on this redundancy, we propose to *forecast in a downsampled-resolution space*. This downsampling-based strategy empowers TSFMs to **see more** historical context by extending the look-back window, and to **forecast better and faster**. It effectively bridges the gap between the need for long-range information and the efficiency constraints of existing TSFMs. However, developing such a downsampling-based approach for enhancing TSFMs faces three challenges:

1. **Information loss.** Naive downsampling leads to significant information loss by indiscriminately removing

[1]Department of Computer Science and Technology, Tsinghua University, Beijing, China [2]ByteDance, Beijing, China [3]Computer Network Information Center, Chinese Academy of Sciences, Beijing, China. Correspondence to: Dan Pei <peidan@tsinghua.edu.cn>.

*Proceedings of the $43^{rd}$ International Conference on Machine Learning*, Seoul, South Korea. PMLR 306, 2026. Copyright 2026 by the author(s).

*Figure 1.* SPRINT is an inference-time play-and-plug framework compatible with any TSFMs. It enhances their capability to accommodate longer look-back windows and generate longer predictions during inference, while achieving higher accuracy, lower costs, and faster efficiency in a training-free way.

high-frequency components, such as spikes and periodic patterns. These information is crucial for accurate forecasting, and their removal compromises the inherent structure of time series.

2. **Resolution Mismatch.** Operating in a downsampled space results in a resolution gap between the coarse-grained downsampled series and the fine-grained target output. Efficient and accurate reconstruction of the fine-grained forecast remains challenging.

3. **Model agnosticism.** For foundational improvement, the approach should act as a universal adapter, allowing seamless integration with any TSFM. Additionally, this integration should require minimal or no modification of the base TSFM to maintain its rapid adaptability across different domains.

To address challenge 1, we propose a framework designed to be explicitly seasonality-preserving, recognizing that seasonal patterns contain significant high-frequency information. Specifically, seasonal-trend decomposition is performed first. To preserve the seasonal component, we employ a robust **Pattern Replication** mechanism, effectively preventing aliasing and information loss caused by downsampling. To address challenge 2, we propose a **Resolution Interpolation** workflow applied selectively to the low-frequency trend, which is more manageable for reconstruction. We feed the downsampled trend component into the base TSFM, allowing it to process a much longer horizon. The TSFM generates a low-resolution prediction, which is then reconstructed back to high resolution via efficient interpolation. Finally, for challenge 3, we propose developing the framework as a **training-free**, plug-and-play wrapper that functions during inference. It views the underlying TSFM as a black box, thereby ensuring compatibility with any architecture without the need for fine-tuning or additional parameter learning.

Ultimately, we propose **SPRINT** (**S**easonal **P**attern **R**eplication and Trend Resolution **INT**erpolation), a novel, training-free, and plug-and-play framework. As shown in Fig 1, it empowers TSFMs to see more historical data and perform long-term and ultra-long-term forecasting accurately and efficiently in a training-free way. The contributions are summarized as follows:

- We are the first to propose a training-free and plug-and-play framework called SPRINT that empowers any existing time series foundation models to overcome scalability bottlenecks during inference, facilitating accurate and efficient long-term and ultra-long-term forecasting.
- In SPRINT, we design a compositional forecasting method for trend and seasonality, including Resolution Interpolation for low-frequency trends and Pattern Replication for high-frequency seasonality.
- We provide both theoretical justifications and in-depth component-wise experiments to validate the effectiveness of SPRINT.
- Extensive experiments on diverse datasets demonstrate that SPRINT not only drastically reduces computational costs but also empowers various TSFMs to achieve new levels of forecasting accuracy.

## 2. Related Work

### 2.1. Long-term Time Series Forecasting

Long-term time series forecasting (LTSF) presents significant challenges due to the inherent high volatility and complex dependencies inherent in long sequences. While numerous approaches with various architectures have been proposed, there are several consensus strategies to tackle these challenges. A key consensus is the crucial role of seasonality modeling. Existing research utilizes strategies such as classic seasonal-trend decomposition (Wu et al., 2021; Zeng et al., 2023; He et al., 2023), explicit seasonality modeling (Lin et al., 2024), and multi-scale seasonality analysis (TimesNet (Wu et al., 2023) and TimeMixer (Wang et al., 2024)). Additionally, another pivotal technique for enhancing accuracy and efficiency is patching, notably developed in PatchTST (Nie et al., 2023). It segments time series into patches, thus preserving local semantic information while significantly reducing computational costs, enabling models to manage longer sequence lengths effectively. In this study, we emphasize the importance of seasonality modeling. Instead of relying on patching, which typically necessitates patch embedding, we utilize training-free downsampling to optimize TSFMs efficiently.

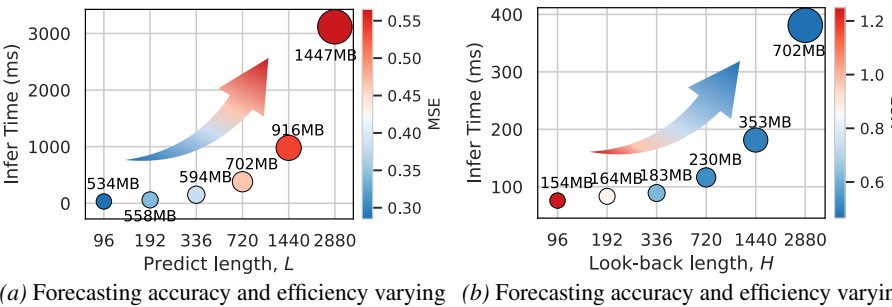

*(a)* Forecasting accuracy and efficiency varying prediction length $L$ with $H = 2880$.

*(b)* Forecasting accuracy and efficiency varying historical look-back $H$ with $L = 720$.

*Figure 2.* Motivational study of the impact of historical look-back lengths ($H$) and prediction lengths ($L$) on forecasting accuracy (MSE), inference time, and max GPU memory. Experiments are conducted on ETTm1 using Moirai with batch size of 12.

### 2.2. Time Series Foundation Models

Currently, the mainstream paradigm for TSFMs leverages the transformer architecture pretrained on large-scale datasets. Based on their backbones, these models are generally classified into three categories. *Decoder-only models* (*e.g.*, TimesFM (Das et al., 2024), Timer (Liu et al., 2024), TimeMoE (Shi et al., 2024) and ToTo (Cohen et al., 2026)) adopt GPT-like architectures for autoregressive forecasting. *Encoder-only models* like Moirai (Woo et al., 2024) focus on learning universal representations via masked modeling objectives. *Encoder-decoder models* such as Chronos (Ansari et al., 2024) treat forecasting as a sequence-to-sequence translation task. *Non-transformer architectures*, notably TTM (Ekambaram et al., 2024), have also emerged but generally lack the ability to support dynamic look-back and prediction lengths. Despite their success, applying these TSFMs to LTSF remains challenging. Their inherent complexity results in significant computational costs and reduced inference efficiency as sequence length grows. Furthermore, zero-shot TSFMs tend to struggle with accurate forecasting in long-term time series. VisionTS (Chen et al., 2025), is a *VLM-based model* that treats time series as images. It faces an accuracy-efficiency trade-off due to its reliance on a fixed image resolution, which might inadequately capture the intricate details of longer sequences.. In this study, we develop a downsampling-based approach to enhance the accuracy and efficiency of TSFMs specifically for LTSF and ultra LTSF.

## 3. Motivation

The inherent complexity of TSFMs often leads to substantial challenges in LTSF, particularly in terms of accuracy, computational costs, and inference efficiency. To understand the limitations of applying TSFMs to LTSF and to motivate our design, we conduct an empirical study using a widely used TSFM called Moirai (2.0 small version) (Woo et al., 2024) on the ETTm1 dataset (Zhou et al., 2021). We analyze the impact of varying the prediction length $L$ and

the historical look-back length $H$ on accuracy (indicated by MSE, see Appendix B.3), computational costs (Max GPU Memory, see Sec. 5.2), and inference efficiency (Infer Time, see Sec. 5.2).

**Observation 1: TSFMs struggle to maintain both accuracy and efficiency when prediction lengths increase.** First, we fix the historical look-back length $H = 2880$ and increase the prediction length $L$. As illustrated in Fig. 2a, extending prediction lengths leads to a marked decline in accuracy (rising MSE) due to the increasing uncertainty and complexity in modeling distant dependency. Simultaneously, both inference time and GPU memory utilization grow significantly as the model predicts longer sequences.

**Observation 2: Increasing look-back lengths help TSFMs for more accurate forecasting but yields higher computational costs and slower inference efficiency.** Since a natural strategy to mitigate the accuracy degradation in LTSF is to provide richer context to capture long-range dependencies, we extend the historical look-back length $H$. As shown in Fig. 2b, increasing $H$ indeed yields a significant improvement in accuracy (lower MSE). However, this accuracy gain comes with a prohibitive cost: the computational complexity of TSFMs typically increases with look-back lengths, leading to a proportional rise in both inference time and GPU memory usage.

To process a long look-back window and predict long future horizon as short series with lower costs and higher efficiency, we propose to *forecast in a downsampled-resolution space*. By reducing temporal resolution, downsampling enables TSFMs to effectively capture long-range dependencies and address a simplified forecasting task, resulting in enhanced accuracy and efficiency.

## 4. Methodology

### 4.1. Problem Definition

Standard time series forecasting is to predict a future time series $\mathbf{Y} = \{\mathbf{x}_{H+1}, \ldots, \mathbf{x}_{H+L}\} \in \mathbb{R}^{L \times C}$ of length $L$ given

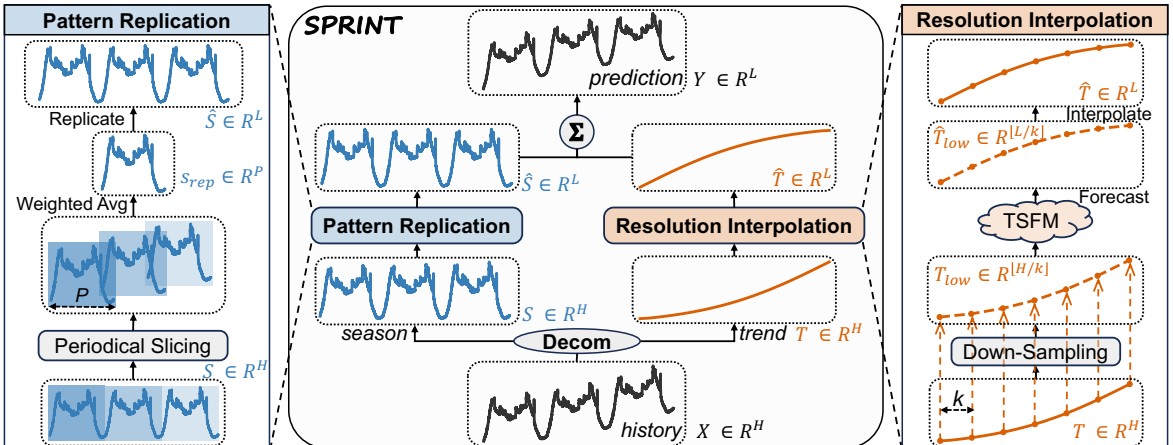

*Figure 3.* Overview of SPRINT (Sec. 4.2). The input series is first decomposed into seasonal and trend components via moving average. The high-frequency seasonal component is predicted using Pattern Replication (left, Sec. 4.4), which robustly repeats recent cycles. The low-frequency trend component is handled by Resolution Interpolation (right, Sec. 4.3), which downsamples the trend for efficient low-resolution forecasting by the base TSFM and then reconstructs the high-resolution trend component via interpolation.

a historical series $\mathbf{X} = \{\mathbf{x}_1, \ldots, \mathbf{x}_H\} \in \mathbb{R}^{H \times C}$ of length $H$, where $C$ is the number of variables (or channels). In this study, we focus on the challenging setting of long-term and ultra long-term time series forecasting for TSFMs, where both $H$ and $L$ are exceptionally large ($H \gg 1, L \gg 1$).

### 4.2. Overall Framework

Based on the idea of "forecast in a downsampled-resolution space", we propose SPRINT (Fig. 3), a training-free, plug-and-play framework that enables existing TSFMs to perform accurate and efficient long-term forecasting. At its core, SPRINT uses moving average (Li et al., 2023) to decompose the input historical series into trend and seasonal components for separate processing. The moving average window size is generally is typically set to match the intrinsic period of the data, denoted as $P$. The low-frequency trend is managed by a Resolution Interpolation workflow to extend the model's effective look-back and prediction lengths while reducing computational costs (Sec. 4.3). Meanwhile, the high-frequency seasonality is preserved with a Pattern Replication mechanism to prevent information loss (Sec. 4.4). Finally, SPRINT generates predictions by summing both trend and seasonal components. Note that SPRINT can be easily integrated into TSFMs during inference without any architectural modifications or additional fine-tuning.

### 4.3. Trend Resolution Interpolation

The primary challenge in LTSF lies in limited sequence lengths of existing TSFMs, which hinder their ability to capture long-range dependencies. To address this, we introduce the **Resolution-Interpolation** workflow, a core component of SPRINT designed specifically for the low-frequency trend component. The core idea is to process a much longer

look-back window by reducing its resolution before forecasting, and then upsampling the TSFM's low-resolution output. This strategy effectively extends the TSFM's receptive field and reduces computational costs without requiring modifications or fine-tuning.

**Low-resolution forecasting.** Firstly, the extracted historical trend component, $\mathbf{T} \in \mathbb{R}^H$, where $H$ denotes the look-back length, is downsampled with an interval $k > 1$. We select every $k$-th point from the series to produce a low-resolution series $\mathbf{T}_{\text{low}} \in \mathbb{R}^{\lfloor H/k \rfloor}$. It is subsequently input into the base TSFM to generate a low-resolution forecast $\hat{\mathbf{T}}_{\text{low}} \in \mathbb{R}^{\lfloor L/k \rfloor}$, where $\lfloor L/k \rfloor$ is the model's prediction length. This method is effective because downsampling retains the essential pattern of the low-frequency trend component. While resolution decreases, the learnability and smooth nature of the trend is preserved. This is detailed in the following proposition (proof in Appendix A.1):

**Proposition 4.1.** *Assume the trend $\mathbf{T}$ follows a smooth temporal process, the downsampled series $\mathbf{T}_{\text{low}}$ preserves this same functional form of temporal process.*

Therefore, a TSFM capable of capturing smooth temporal patterns can effectively predict $\mathbf{T}_{\text{low}}$ without fine-tuning.

**High-resolution reconstruction.** Secondly, the low-resolution forecast $\hat{\mathbf{T}}_{\text{low}}$ is reconstructed back to the original high resolution. This reconstruction is based on the fundamental sampling theorem (Shannon, 2006):

**Theorem 4.2** (Nyquist-Shannon Sampling Theorem). *A band-limited continuous signal can be perfectly reconstructed from its discrete samples if the sampling rate exceeds twice the maximum frequency of the signal.*

In SPRINT, the trend is derived using moving average with a window size $P$, functioning as a low-pass filter with a cutoff

frequency of $f_{\max} \approx 1/P$. To satisfy the Nyquist criterion (the sampling rate $1/k > 2f_{\max}$), we require $k < P/2$. Given that data period typically satisfies $P \gg 2$ in real-world scenarios, such as $P = 24$ for hourly data, this requirement is met for a broad range of $k$, e.g., $P/3$ and $P/4$. For resolution reconstruction, we employ a continuous *interpolation* technique, e.g., spline or linear interpolation. Unlike discontinuous techniques like nearest-neighbor interpolation, which may introduce artificial sharp angles, continuous interpolation ensures the resulting curve is smooth. This smoothness is particularly advantageous for reconstructing the slowly-varying trend component. By default, SPRINT employs spline interpolation. Given low-resolution forecast $(\tau_i, \hat{T}_{\text{low},i})$, where $\tau_i$ are the coarse time indices, we construct a spline function $S(t)$. By evaluating this function at the original high-resolution time indices $\{0, 1, \ldots, L-1\}$, we obtain the high-resolution forecast $\hat{T} \in \mathbb{R}^L$.

### 4.4. Seasonal Pattern Replication

While the Resolution-Interpolation workflow efficiently processes the low-frequency trend component, it is not suitable for the entire time series. High-frequency seasonal components contain crucial information such as spikes and fluctuating patterns. Downsampling would lose these high-frequency information, leading to significant forecasting inaccuracies. To address this, we propose the **Pattern Replication** mechanism, a simple yet effective approach that preserves critical high-frequency information. The core idea is not to downsample the seasonal component, but to explicitly extract a representative seasonal pattern from historical sequence and replicate it into the forecast horizon.

Specifically, from the historical seasonal component $\mathbf{S} \in \mathbb{R}^H$, we extract the last $N = \lfloor H/P \rfloor$ complete seasonal cycles, denoted as $\mathbf{s}_1, \mathbf{s}_2, \ldots, \mathbf{s}_N \in \mathbb{R}^P$, where $P$ is the inherent dominant period of the data and $\mathbf{s}_N$ is the most recent cycle. Instead of using only the latest cycle, which may be affected by noise, we compute a more robust representative pattern, $\mathbf{s}_{\text{rep}} \in \mathbb{R}^P$, using an exponential weighted average of these historical cycles. Therefore, $\mathbf{s}_{\text{rep}}$ integrates comprehensive historical information while giving higher importance to more recent data. It is calculated as $\mathbf{s}_{\text{rep}} = (\sum_{i=1}^N w^{N-i} \mathbf{s}_i)/\sum_{i=1}^N w^{N-i}$, where $w \in (0,1]$ serves as the weighting base. This formulation ensures that the most recent cycle $\mathbf{s}_N$ is assigned the highest weight, proportional to $w^0 = 1$, while earlier cycles receive exponentially decaying weights. The future seasonal forecast, $\hat{S} \in \mathbb{R}^L$, is then generated by simply repeating this robust pattern throughout the forecast horizon $L$. Formally, the predicted value at a future time point $t$ is given by:

$$\hat{S}_t = s_{\text{rep},t \pmod P}, t = 0, 1, \ldots, L-1$$

As demonstrated in the following proposition (proof in Appendix A.2), the error of pattern replication shows only polynomial growth, thus avoiding the exponential error accumulation commonly seen in autoregressive forecasting.

**Proposition 4.3.** *Assume the seasonal component* $\mathbf{S}$ *is quasi-periodic with period* $P$, $S_t = S_{t-P} + \epsilon_t$. *The cumulative error of Pattern Replication with a prediction length* $L$ *is bounded by* $O(L^2 \epsilon_{\max})$.

## 5. Experiments

In this section, we present the superior forecasting accuracy and exceptional efficiency of SPRINT. To demonstrate its practical plug-and-play design, we integrate SPRINT into several state-of-the-art and widely recognized TSFMs with various architectures. The *encoder-decoder* models include Chronos (Small) (Ansari et al., 2024). Among the *encoder-only* models, we include Moirai (2.0 Small) (Woo et al., 2024). For the mainstream *decoder-only* models, we cover TimesFM (Das et al., 2024), TimeMoE (Shi et al., 2024), Timer (Liu et al., 2024), and ToTo (Cohen et al., 2026). We also assess a *VLM-based* model called VisionTS (Chen et al., 2025). *Non-transformer architectures*, such as TTM, are excluded since they cannot support dynamic look-back and forecasting lengths. Additionally, we provide results of the simple but classic Seasonal Naive method as a baseline.

Our evaluation spans 9 datasets across 5 key domains, of which 8 are standard benchmarks. We also introduce a new dataset, Service, characterized by high sampling rates, long-range dependencies, and pronounced high-frequency dynamics. We report the standard Mean Squared Error (MSE) to compare TSFM performance both with and without SPRINT. All experiments are conducted in a **zero-shot** setting, aligning with TSFM's objective of rapid adaptation to diverse domains. For SPRINT's configuration, we set the downsampling interval $k = P/4$, the seasonal decay factor $w = 0.9$, and the look-back length $H = 30P$ by default, where $P$ is the intrinsic period of each dataset. Detailed descriptions of datasets, TSFMs, evaluation metrics, and implementation are provided in Appendix B. Due to space limit, analysis of parameter sensitivity, interpolation methods and probabilistic forecasting is presented in Appendix F, D and G.

### 5.1. Accuracy Analysis

Tbl. 1 summarizes the forecasting accuracy for standard LTSF, covering prediction lengths $L \in \{96, 192, 336, 720\}$ across all datasets and TSFMs. SPRINT consistently enhances accuracy for nearly all backbone models. To quantify this improvement, we calculate the average relative MSE reduction across all datasets for each TSFM. SPRINT achieves significant average reductions ranging from 6.60% to 29.81% across the evaluated TSFMs. These gains are particularly pronounced on high-frequency datasets with strong seasonality, such as Solar and Service. Overall, these results

*Table 1.* Results of LTSF, which are averaged from all prediction lengths of $L \in \{96, 192, 336, 720\}$. **Bold** indicates best result. Underlined indicates worse than Seasonal Naive. A slash (/) denotes datasets included in the model's pretraining and thus excluded from testing. Detailed results can be found in Tbl. 11.

| Dataset | Seasonal Naive | Encoder-Decoder | | Encoder-only | | Decoder-only | | | | | | | | VLM-based | |
|---|---|---|---|---|---|---|---|---|---|---|---|---|---|---|---|
| | | Chronos | +SPRINT | Moirai | +SPRINT | TimesFM | +SPRINT | TimeMoE | +SPRINT | Timer | +SPRINT | ToTo | +SPRINT | VisionTS | +SPRINT |
| ETTh1 | 0.600 | 0.527 | **0.465** | 0.446 | **0.410** | 0.479 | **0.422** | 0.409 | **0.406** | 0.412 | **0.396** | 0.441 | **0.395** | 0.778 | **0.492** |
| ETTh2 | 0.483 | 0.383 | **0.363** | 0.372 | **0.354** | 0.402 | **0.371** | 0.342 | **0.338** | 0.356 | **0.343** | 0.370 | **0.331** | 0.397 | **0.367** |
| ETTm1 | 0.489 | 0.615 | **0.414** | 0.369 | **0.356** | 0.429 | **0.359** | 0.378 | **0.346** | 0.395 | **0.357** | 0.380 | **0.347** | 0.709 | **0.452** |
| ETTm2 | 0.358 | 0.321 | **0.260** | 0.286 | **0.264** | 0.332 | **0.278** | 0.286 | **0.246** | 0.279 | **0.254** | 0.263 | **0.246** | 0.320 | **0.284** |
| Weather | 0.371 | 0.288 | **0.239** | 0.254 | **0.238** | / | / | 0.383 | **0.220** | 0.243 | **0.227** | 0.218 | **0.217** | 0.290 | **0.266** |
| Solar | 0.341 | 0.876 | **0.255** | **0.221** | 0.230 | 0.450 | **0.212** | 0.513 | **0.204** | 0.524 | **0.212** | 0.260 | **0.215** | 0.745 | **0.349** |
| Wind | 1.869 | 1.367 | **1.185** | 1.187 | **1.170** | 1.613 | **1.099** | 1.099 | **1.072** | 1.148 | **1.091** | 1.171 | **1.100** | 1.210 | **1.161** |
| ZafNoo | 0.853 | 0.633 | **0.546** | 0.583 | **0.555** | 0.633 | **0.537** | 0.534 | **0.508** | 0.559 | **0.520** | 0.537 | **0.524** | 0.939 | **0.542** |
| Service | 0.280 | 1.006 | **0.154** | 0.304 | **0.161** | 0.954 | **0.157** | 0.575 | **0.153** | 0.745 | **0.494** | **0.166** | 0.171 | 0.426 | **0.491** |
| **Avg. Improv.** | - | - | **29.81%** | - | **8.85%** | - | **29.46%** | - | **23.10%** | - | **15.32%** | - | **6.60%** | - | **20.48%** |

empirically validate SPRINT as a universal and effective enhancement for diverse TSFMs in standard LTSF.

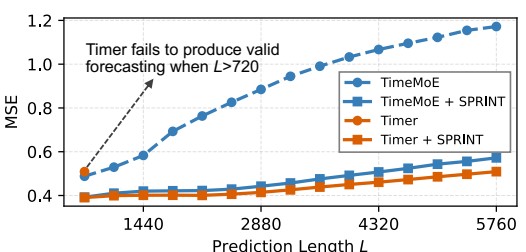

*Figure 4.* Accuracy comparison of TSFMs with and without SPRINT with varying prediction length $L$. Using TimeMoE and Timer as backbones on ETTm1, SPRINT consistently reduces errors and enables Timer to function beyond its native failure length.

Furthermore, in ultra LTSF with $L = 1440$ (detailed results in Appendix C), SPRINT demonstrates a pronounced performance advantage. Specifically, SPRINT yields average MSE reductions ranging from 18.45% to 32.39% across the evaluated TSFMs. Fig. 4 illustrates the MSE trends as the prediction length $L$ extends from 720 to 5760 on ETTm1, utilizing TimeMoE and Timer as backbones. The results indicate that SPRINT consistently reduces forecasting errors across all extended horizons for base TSFMs. Notably, Timer without SPRINT fails to produce valid predictions (resulting in NaN) when $L > 720$. However, equipping it with SPRINT successfully resolves this issue, allowing Timer to function robustly even at $L = 5760$. This demonstrates SPRINT's robustness and scalability in handling extended forecast horizons where standard TSFMs typically struggle.

### 5.2. Efficiency Analysis

Beyond achieving superior forecasting accuracy, SPRINT demonstrates remarkable computational efficiency as a plug-and-play enhancement for TSFMs. We conduct an efficiency analysis and examine four key metrics: (1) **Parameters**: the total count of model parameters; (2) **MACs (Multiply-Accumulate Operations)**: the computational complexity representing the number of multiply-accumulate operations per sample; (3) **Max Memory**: maximum GPU memory

*Table 2.* Model efficiency comparison on ETTm1, with look-back length $H = 2880$ and prediction length $L = 720$. MACs here indicate the computational cost for processing a single sample, and max memory usage is measured with a batch size of 12.

| Model | Parameters | MACs | Max Mem | Infer Time (ms) |
|---|---|---|---|---|
| Chronos | 46.15M | 6.44T | 12.48GB | 1707.72 |
| **+SPRINT** | - | 181.92G | 3.40GB | 28.45 |
| Moirai | 11.39M | 3.59T | 701.71MB | 381.14 |
| **+SPRINT** | - | 17.05G | 82.85MB | 7.19 |
| TimesFM | 203.57M | 627.81G | 827.03MB | 17.38 |
| **+SPRINT** | - | 104.63G | 815.62MB | 3.95 |
| TimeMoE | 453.20M | 4.99T | 42.95GB | 321.76 |
| **+SPRINT** | - | 166.65G | 2.93GB | 15.95 |
| Timer | 84.14M | 21.79G | 804.03MB | 3.83 |
| **+SPRINT** | - | 589.03M | 420.96MB | 1.42 |
| ToTo | 151.31M | 6.11T | 12.67GB | 906.45 |
| **+SPRINT** | - | 218.37G | 1.21GB | 33.52 |
| VisionTS | 55.81K | 50.07G | 901.23MB | 6.43 |
| **+SPRINT** | - | 50.07G | 901.50MB | 7.33 |
| *Reduction* | - | 343.62× ↓ | 6.36× ↓ | 16.87× ↓ |

usage during inference; and (4) **Infer Time**: average inference time per sample on GPU. Notably, since SPRINT operates as a preprocessing-postprocessing wrapper *outside* the TSFM, the parameter count remains unchanged.

**Main efficiency comparison.** We firstly conduct experiments with look-back length $H = 30P = 30 \times 96 = 2880$ for ETTm1, prediction length $L = 720$, and a batch size of 12. Tbl. 2 compares the efficiency of TSFMs with and without SPRINT. The results reveal significant computational savings for most TSFMs. Specifically, SPRINT achieves an average reduction of **343.62×** in MACs, **6.35×** in max memory usage, and **16.87×** in inference latency compared to the base models. These efficiency gains arise from the reduced sequence length fed into the base TSFM, as computational complexity in attention-based architectures typically scales quadratically with sequence length. Crucially, these efficiency improvements are also associated with enhanced forecasting accuracy, as shown in Tbl. 1 and Tbl. 7. Note that the integration of SPRINT into VisionTS yields no efficiency improvement since it treats time series of various

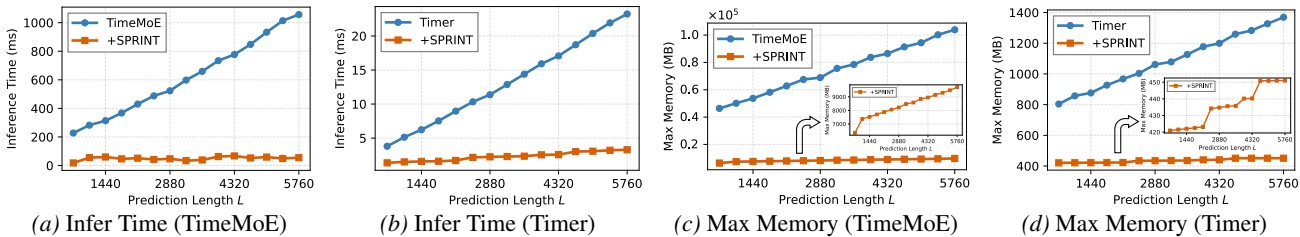

*Figure 5.* Efficiency comparison of TSFMs with and without SPRINT with varying prediction length $L$. Using TimeMoE and Timer as backbones on ETTm1, we fix the historical length at $H = 2880$ and batch size at 12.

lengths as a fixed-resolution image. Overall, the advantages of SPRINT in both efficiency and accuracy make it well-suited for resource-constrained environments.

**Efficiency with longer prediction lengths.** To further assess the scalability of SPRINT, we conduct experiments under ultra-long prediction lengths. specifically, we fix the look-back length at $H = 2880$ and the batch size at 12, while varying the prediction length $L$ from 720 to 5760. We focus on two TSFMs with different sizes (TimeMoE and Timer) and two critical efficiency metrics (Max Memory and Infer Time). As illustrated in Fig. 5, standard TSFMs typically exhibit a sharp increase in computational costs as prediction lengths increase. In contrast, SPRINT consistently maintains a low computational cost. Even as the prediction length extends several times, the expansion in memory usage and inference time for SPRINT-enhanced models remains minimal compared to the base TSFMs. This exceptional scalability is due to the resolution-interpolation mechanism: the underlying TSFM only needs to forecast a downsampled sequence of length $L/k$, thereby avoiding the prohibitive computational costs of long-range forecasting. Overall, SPRINT effectively unlocks the capability of existing TSFMs to handle ultra-long forecasting without carrying great overheads.

### 5.3. Ablation Study

To verify the contribution of each component in SPRINT, we conduct ablation studies on 4 datasets with different sampling rates (ETTh2, ETTm2, Weather, and Service) using 3 representative TSFMs: Chronos (encoder-decoder), Moirai (encoder-only), and TimeMoE (decoder-only). We compare the full SPRINT framework against five variants: (1) **w/o Decomposition**: applying the Resolution-Interpolation workflow directly to the raw series without separating seasonality and trend; (2) **STL Decomposition**: replacing moving average decomposition with STL decomposition (Cleveland et al., 1990); (3) **Season Only**: predicting solely via seasonal pattern replication without modeling the trend; (4) **Season Last**: using the last observed cycle directly for the representative seasonal pattern, $\mathbf{s}_{rep}$, instead of using weighted average; and (5) **w/o Trend Downsampling**: predicting the trend component at the original resolution

without downsampling. We also conduct an analysis of different interpolation methods of the Resolution-Interpolation workflow in Appendix D.

The comparative results presented in Fig. 6 demonstrate that the complete SPRINT consistently outperforms all ablated variants. We analyze each variant's impact: SPRINT **w/o Decomposition** leads to a significant performance decline, confirming that applying downsampling to the raw time series loses high-frequency seasonal details. SPRINT **STL Decomposition** performs worse than moving average decomposition and incurs higher computational costs, as STL does not support tensor computation. On the Service dataset, the long look-back window results in unacceptable runtime. SPRINT **Season Only** produces large errors, highlighting the necessity of modeling the trend component to capture non-stationary drifts. SPRINT **Season Last** is less effective than weighted average for the Pattern Replication module, emphasizing the significance of utilizing global context to address pattern shifts and local noise. SPRINT **w/o Trend Downsampling** also increases forecasting errors, indicating that downsampling can effectively mitigate noise and simplify forecasting. Overall, each component within SPRINT plays a vital role in enhancing its overall effectiveness.

## 6. Component-wise Experimental Analysis

The superior performance of SPRINT relies on decomposition and forecasting with different components. To provide deeper insights into the rationale behind SPRINT, we conduct in-depth component-wise experiments to answer 3 fundamental questions:

1. Can the Pattern Replication mechanism sufficiently predict the high-frequency seasonal component?
2. Can existing TSFMs effectively forecast the trend component in a downsampled-resolution space?
3. Can the high-resolution trend be accurately recovered from the low-resolution output via spline interpolation?

### 6.1. Q1: Seasonal Prediction via Pattern Replication

We first evaluate the effectiveness of the **Pattern Replication** mechanism for seasonal prediction. Unlike trends,

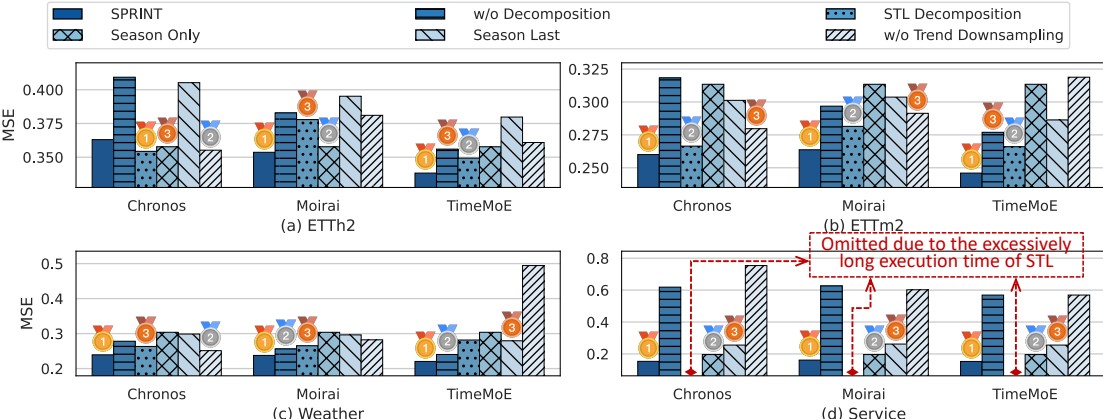

*Figure 6.* Ablation study results on ETTh2, ETTm2, Weather, and Service datasets using Chronos, Moirai, and TimeMoE backbones.

*Table 3.* Accuracy of different components of SPRINT. For trend interpolation, the target length is set to $30P$ with a sampling interval of $P/4$. For other components, MSE is averaged over $L \in \{96, 192, 336, 720\}$.

| Dataset | Season Forecast | Trend Interp. | Trend (Forecast with downsampled series) | | | | | | |
|---|---|---|---|---|---|---|---|---|---|
| | | | Chronos | Moirai | TimesFM | TimeMoE | Timer | ToTo | VisionTS |
| ETTh1 | 0.130 | **0.004** | 0.274 | 0.220 | 0.234 | 0.214 | 0.205 | 0.204 | 0.312 |
| ETTh2 | 0.051 | **0.001** | 0.297 | 0.287 | 0.304 | 0.272 | 0.277 | 0.263 | 0.302 |
| ETTm1 | 0.121 | **0.004** | 0.244 | 0.189 | 0.194 | 0.179 | 0.188 | 0.179 | 0.296 |
| ETTm2 | 0.049 | **0.001** | 0.192 | 0.195 | 0.209 | 0.177 | 0.185 | 0.174 | 0.217 |
| Weather | 0.066 | **0.001** | 0.160 | 0.158 | 0.155 | 0.141 | 0.147 | 0.136 | 0.188 |
| Solar | 0.087 | **0.003** | 0.162 | 0.137 | 0.121 | 0.110 | 0.122 | 0.119 | 0.281 |
| Wind | 0.196 | **0.004** | 0.856 | 0.840 | 0.777 | 0.743 | 0.764 | 0.759 | 0.842 |
| ZafNoo | 0.146 | **0.002** | 0.373 | 0.381 | 0.364 | 0.335 | 0.347 | 0.344 | 0.370 |

seasonal components contains high-frequency pattern but are inherently repetitive. Based on the theoretical insights from Proposition 4.3, we suggest that future seasonality in time series can be reliably predicted via pattern replication, without the need for complex deep learning-based techniques. To validate this, we extract the ground-truth seasonal components from the test set of each dataset using moving average. We then generate predictions solely using the Pattern Replication module, reporting MSE between the replicated pattern and the actual future seasonal component.

As illustrated in the "Season Forecast" column of Tbl. 3, the simple approach achieves remarkably lower errors than the overall forecasting errors (Tbl. 1). The results align well with the theoretical bounds in Proposition 4.3. Together, they demonstrate the feasibility of bypassing TSFMs for seasonal forecasting, thereby conserving computational resources. Visual examples of the replicated seasonal patterns versus ground truth are provided in Appendix J.2.

### 6.2. Q2: Trend Prediction with Low Resolution

A core premise of the **Resolution Interpolation** workflow is that existing TSFMs can effectively model the trend component after it has been downsampled. Based on the theoretical guarantee in Proposition 4.1, we hypothesize that downsam-

pling alleviates complexity by removing high-frequency noise and extending the receptive field, allowing TSFMs to better capture long-term dependencies. To verify this, trend components are extracted from the test datasets using moving average. We then downsample these trends and input them into the base TSFMs to generate predictions, computing errors against the downsampled ground-truth trend.

Tbl. 3 reports the MSE of trend forecasts using downsampled series. Consistent with Proposition 4.1, the errors of forecasting downsampled trend are significantly lower than overall forecasting errors (Tbl. 1). This indicates that the "coarse-grained" view provided by downsampling preserves essential dynamics for long-term forecasting, confirming that base TSFMs are proficient at modeling the simplified trajectories. Visualizations of the trend forecasts can be found in Appendix J.3.

### 6.3. Q3: Trend Reconstruction via Interpolation

Finally, we investigate the information loss during the reconstruction process, which is grounded in the Nyquist-Shannon Sampling Theorem (Theorem 4.2). Given that the trend component is inherently low-frequency, we assume it can be accurately recovered from low-resolution data via spline interpolation. To validate this, we conduct a reconstruction experiment. We take the ground-truth trend series, downsample it by a factor of k = P/4, and then immediately upsample it back to the original resolution using spline interpolation, reporting reconstruction MSE between the reconstructed trend and the original ground-truth trend.

As shown in the "Trend Interp." column of Tbl. 3, the reconstruction errors are negligible compared to errors of overall forecasting (Tbl. 1) and other components. The result aligns with Theorem 4.2 and confirms that the trend component's slow-varying nature allows for aggressive downsampling with minimal information loss. Additional showcases com-

paring original and reconstructed trends are provided in Appendix J.4.

## 7. Applicability to Non-periodic Data

The default design of SPRINT operates under the assumption that a time series exhibits strong seasonality. When seasonality is weak or unstable, directly replicating seasonal patterns may be unreliable. In such scenarios, SPRINT can fall back to the Resolution-Interpolation workflow without seasonal-trend decomposition. Specifically, non-seasonal series are downsampled and predicted in the low-resolution space, with the downsampling interval determined by the specific characteristics of the data.

**Automatic $k$-selection.** To automatically determine the downsampling interval $k$ for non-periodic data, we propose selecting $k$ based on the *smoothness* of the input data. The smoothness of each input series is quantified as

$$s = r_{\text{turn}} \cdot v_{\text{norm}}$$

, where $r_{\text{turn}}$ denotes the *turning points ratio* and $v_{\text{norm}}$ denotes the *normalized volatility*. The turning points ratio quantifies the proportion of local extrema, reflecting how frequently the direction of a series changes. The normalized volatility measures the average adjacent-step fluctuation after scale normalization. A smaller $s$ therefore indicates smoother dynamics. We use this score to guide the selection of the downsampling interval $k$: smoother data allow for larger $k$, whereas highly volatile data trigger $k = 1$, which bypasses downsampling.

*Table 4.* Relationship between smoothness ($s$) and the optimal downsampling interval ($k_{\text{opt}}$) on non-periodic financial datasets. 'S2', 'S4', 'S8', and 'S16' denote variants smoothed by moving average windows of corresponding sizes.

| Dataset | Variant | Smooth. $s\ (10^{-3})\downarrow$ | $k_{\text{opt}}$ | MSE |
|---|---|---|---|---|
| Exchange | Raw | 10.5 | 1 | 0.0395 |
| | S2 | 5.60 | 2 | 0.0379 |
| | S4 | 2.85 | 2 | 0.0386 |
| | S8 | 1.38 | 4 | 0.0396 |
| | S16 | 0.720 | 8 | 0.0396 |
| NASDAQ | Raw | 12.3 | 1 | 1.040 |
| | S2 | 7.12 | 1 | 1.080 |
| | S4 | 4.36 | 4 | 1.096 |
| | S8 | 2.62 | 4 | 0.955 |
| | S16 | 1.50 | 4 | 0.829 |
| NYSE | Raw | 6.15 | 4 | 0.637 |
| | S2 | 3.73 | 4 | 0.638 |
| | S4 | 2.33 | 4 | 0.639 |
| | S8 | 1.40 | 8 | 0.644 |
| | S16 | 0.853 | 8 | 0.599 |

**Experimental setup.** We validate the selection method on 3 day-sampled, non-periodic financial datasets: Exchange (Lai et al., 2018), NASDAQ (Feng et al., 2019),

and NYSE (Feng et al., 2019). The look-back length is set to $H = 64$, and MSE is averaged over prediction lengths $L \in \{24, 36, 48, 60\}$. To obtain series with different smoothness levels, we additionally create smoothed variants by applying moving average to the raw data (*e.g.*, S4 represents a moving average window of size 4). For each dataset variant, we sweep candidate downsampling intervals and report the optimal interval $k_{\text{opt}}$ selected by the lowest MSE.

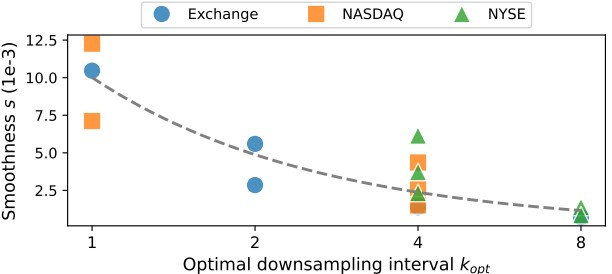

*Figure 7.* Smoothness ($s$) vs. the optimal downsampling interval ($k_{\text{opt}}$) on non-periodic financial datasets, including their various smoothed variants. A lower $s$ corresponds to smoother data, allowing for larger downsampling intervals.

As illustrated in Tbl. 4 and Fig. 7, smoother variants consistently prefer larger downsampling intervals. Across all datasets and variants, the Pearson correlation between the smoothness score ($s$) and $\log_2(k_{\text{opt}})$ is $-0.859$, indicating a strong monotonic relationship between intrinsic smoothness and the downsampling intervals. Thus, SPRINT can automatically route non-seasonal data to an appropriate resolution using a lightweight smoothness threshold.

## 8. Conclusion

In this paper, we introduce SPRINT, a novel, training-free, and play-and-plug framework designed to overcome the accuracy and efficiency bottlenecks of TSFMs in long-term and ultra-long-term forecasting. Based on the idea of "forecast in a downsampled-resolution space", SPRINT effectively extends the receptive field of base TSFMs while reducing computational costs. Specifically, we design a Resolution Interpolation workflow to enable TSFMs to capture long-term low-frequency trends from downsampled series, and a Pattern Replication mechanism to robustly preserve high-frequency seasonal details without information loss. Supported by theoretical guarantees and extensive experimental results, SPRINT is proven to significantly boost both the accuracy and efficiency of existing TSFMs, offering a robust solution for scalable long-term forecasting. Detailed discussions on the limitations and directions for future research are provided in Appendix H.

## Acknowledgements

This work is supported by the National Key Research and Development Program of China (No.2024YFB4505903).

## Impact Statement

This work advances the field of Machine Learning, specifically by enhancing the accuracy and efficiency of time series foundation models. The primary goal is to overcome the length limitations of current foundation models to enable accurate long-term forecasting with reduced computational costs. There are many potential societal consequences of our work, none of which we feel must be specifically highlighted here.

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

# A. Theoretical Analysis of SPRINT

## A.1. Theoretical Analysis of Trend Resolution Interpolation

### A.1.1. PROOF OF PROPOSITION 4.1

To validate Proposition 4.1, we assume the trend component as a first-order autoregressive process (AR(1)). Our proof illustrates that downsampling an AR(1) process yields another AR(1) process, thereby maintaining the fundamental structure of the trend series.

**Assumption.** Let the original high-resolution trend series $\{x_t\}$ be generated by a stationary AR(1) process:

$$x_t = \phi x_{t-1} + \epsilon_t, \quad \epsilon_t \sim \mathcal{N}(0, \sigma^2) \tag{1}$$

where $\phi$ is the autoregressive coefficient, and $\epsilon_t$ is the irregular noise. For a trend component, which is typically smooth and slowly varying, the coefficient $\phi$ is close to 1.

We apply a downsampling operation with interval $k$, selecting every $k$-th data point to form the low-resolution series $\{y_\tau\}$, where $y_\tau = x_{k\tau}$. We analyze the relationship between successive points $y_\tau$ and $y_{\tau-1}$, *i.e.*, $x_{k\tau}$ and $x_{k(\tau-1)}$.

By recursive substitution of the AR(1) equation $k$ times:

$$
\begin{aligned}
x_t &= \phi x_{t-1} + \epsilon_t \\
&= \phi(\phi x_{t-2} + \epsilon_{t-1}) + \epsilon_t = \phi^2 x_{t-2} + \phi \epsilon_{t-1} + \epsilon_t \\
&\vdots \\
&= \phi^k x_{t-k} + \sum_{j=0}^{k-1} \phi^j \epsilon_{t-j}
\end{aligned}
\tag{2}
$$

Substituting $t = k\tau$, we obtain the governing equation for the downsampled series $y_\tau$:

$$y_\tau = \phi^k y_{\tau-1} + \eta_\tau \tag{3}$$

where $\eta_\tau = \sum_{j=0}^{k-1} \phi^j \epsilon_{k\tau-j}$ is the aggregated noise term.

The derived equation $y_\tau = \phi^k y_{\tau-1} + \eta_\tau$ demonstrates that the downsampled series $\{y_\tau\}$ is mathematically equivalent to a new AR(1) process with:

- **New Coefficient:** $\phi' = \phi^k$. For a stationary process, $|\phi| < 1$, hence $|\phi'| < 1$, which maintains stationarity. When the $\phi$ of a trend series satisfies $\phi \approx 1$, $\phi^k$ remains close to 1, resulting in a series that is smooth and slowly varying.
- **New Noise Variance:** The variance of the new noise $\eta_\tau$ is given by $\text{Var}(\eta_\tau) = \sigma^2 \sum_{j=0}^{k-1} (\phi^2)^j = \sigma^2 \frac{1-\phi^{2k}}{1-\phi^2}$.

Through downsampling, the original AR(1) process is transformed into another AR(1) process, preserving the linear dependency on the previous state. When general TSFMs can capture such fundamental temporal dynamics, they can readily generalize to the downsampled series $\mathbf{T}_{\text{low}}$. Importantly, this implies that SPRINT enables TSFMs to perdorm zero-shot forecasting on the downsampled trend without requiring to learn new, complex dynamics, as the underlying temporal process remains invariant.

### A.1.2. THEOREM 4.2

Theorem 4.2 in Section 4.3 refers to the **Nyquist-Shannon Sampling Theorem**. This is a fundamental principle in information theory and signal processing. In the context of our Resolution-Interpolation workflow, this theorem provides the theoretical guarantee that the downsampling does not irreversibly destroy information. Specifically, since the trend is obtained via a moving average filter with window $P$, its frequency content is concentrated below $1/P$. Therefore, satisfying the reconstruction condition merely requires the downsampling interval $k < P/2$. We construct the interpolated result using spline interpolation, which serves as a practical, high-quality approximation to the ideal sinc reconstruction formula implied by the theorem.

## A.2. Theoretical Analysis of Seasonal Pattern Replication

### A.2.1. PROOF OF PROPOSITION 4.3

Let the forecast start at time $t_0$ for a prediction length of $L$. We aim to bound the cumulative error $\sum_{j=1}^{L} |e_j|$, where $e_j = S_{t_0+j} - \hat{S}_{t_0+j}$.

**Assumption (Quasi-Periodicity).** The seasonal component $S_t$ satisfies:

$$S_t = S_{t-P} + \epsilon_t, \quad |\epsilon_t| \leq \epsilon_{\max}, \quad \mathbb{E}[\epsilon_t] = 0 \tag{4}$$

where $P$ is the period and $\epsilon_t$ is a bounded noise.

The Pattern Replication forecast, denoted as $\hat{S}_{t_0+j}$, employs a representative pattern extracted from historical cycles. For illustration, consider the scenario where it replicates the most recent cycle, which constitutes a component of the weighted average:

$$\hat{S}_{t_0+j} = S_{t_0+j-kP} \tag{5}$$

where $k = \lceil j/P \rceil$ signifies the number of cycles in the past necessary to locate the corresponding known value. Regarding the weighted average in the Pattern Replication mechanism of SPRINT, we maintain that it does not impact our subsequent proof. Moreover, the weighted average enhances the robustness of the representative pattern in real-world time series by accommodating pattern shifts and local noise.

By recursively applying the quasi-periodic definition $k$ times:

$$\begin{aligned}
S_{t_0+j} &= S_{t_0+j-P} + \epsilon_{t_0+j} \\
&= S_{t_0+j-2P} + \epsilon_{t_0+j-P} + \epsilon_{t_0+j} \\
&\vdots \\
&= S_{t_0+j-kP} + \sum_{m=0}^{k-1} \epsilon_{t_0+j-mP}
\end{aligned} \tag{6}$$

Since $\hat{S}_{t_0+j} = S_{t_0+j-kP}$, the error at time index $j$ is the sum of noise:

$$e_j = S_{t_0+j} - \hat{S}_{t_0+j} = \sum_{m=0}^{k-1} \epsilon_{t_0+j-mP} \tag{7}$$

The error at time index $j$ is bounded by:

$$|e_j| \leq \sum_{m=0}^{k-1} |\epsilon_{t_0+j-mP}| \leq k \cdot \epsilon_{\max} \approx \frac{j}{P} \epsilon_{\max} \tag{8}$$

Finally, we sum the error bounds over the full forecasting horizon $L$:

$$\sum_{j=1}^{L} |e_j| \leq \sum_{j=1}^{L} \frac{j}{P} \epsilon_{\max} = \frac{\epsilon_{\max}}{P} \sum_{j=1}^{L} j = \frac{\epsilon_{\max}}{P} \frac{L(L+1)}{2} = O(L^2 \epsilon_{\max})$$

This result verifies that the error accumulates at a polynomial rate ($L^2$), which is considerably more stable than the exponential error growth ($e^{\lambda L}$) typical of autoregressive forecasting.

# B. Experimental Details

## B.1. Foundation Models

We evaluate SPRINT alongside several recent and well-known (zero-shot) TSFMs for time series: Chronos (small) (Ansari et al., 2024), Moirai (2.0 small) (Woo et al., 2024), TimesFM (Das et al., 2024), TimeMoE (Shi et al., 2024), Timer (Liu et al., 2024), ToTo (Cohen et al., 2026) and VisionTS (Chen et al., 2025). We ignore TTM since it cannot support dynamic look-back and forecasting horizons. For each TSFM, we report results both with and without SPRINT applied. Detailed information of these models is provided in Tbl. 5.

*Table 5.* Detailed information of time series foundation models used in experiments.

| Model | Architecture | Model Size | Download Link |
|---|---|---|---|
| Chronos | encoder-decoder | 46M | https://huggingface.co/amazon/chronos-t5-small |
| Moirai | encoder-only | 11.4M | https://huggingface.co/Salesforce/moirai-2.0-R-small |
| TimesFM | decoder-only | 200M | https://huggingface.co/google/timesfm-1.0-200m |
| TimeMoE | decoder-only | 453M | https://huggingface.co/Maple728/TimeMoE-200M |
| Timer | decoder-only | 84M | https://huggingface.co/thuml/timer-base-84m |
| ToTo | decoder-only | 151M | https://huggingface.co/Datadog/Toto-Open-Base-1.0 |
| VisionTS | VLM-based | 55.8K | https://github.com/Keytoyze/VisionTS |

## B.2. Datasets

*Table 6.* Statistics of datasets

| Dataset | Frequency | Lengths | Dim | Split | Cycle $P$ | Hist Len $T$ |
|---|---|---|---|---|---|---|
| ETTh1 (Zhou et al., 2021) | 1 hour | 14,400 | 7 | 6:2:2 | 24 | 720 |
| ETTh2 (Zhou et al., 2021) | 1 hour | 14,400 | 7 | 6:2:2 | 24 | 720 |
| ETTm1 (Zhou et al., 2021) | 15 mins | 57,600 | 7 | 6:2:2 | 96 | 2880 |
| ETTm2 (Zhou et al., 2021) | 15 mins | 57,600 | 7 | 6:2:2 | 96 | 2880 |
| Weather (Wu et al., 2021) | 10 mins | 52,696 | 21 | 7:1:2 | 144 | 4320 |
| Solar (Lai et al., 2018) | 10 mins | 52,560 | 137 | 6:2:2 | 144 | 4320 |
| Wind (Li et al., 2022) | 15 mins | 48,673 | 7 | 7:1:2 | 96 | 2880 |
| ZafNoo (Poyatos et al., 2020) | 30 mins | 19,225 | 11 | 7:1:2 | 48 | 1440 |
| Service (self-collected) | 1 min | 40,805 | 20 | 7:1:2 | 1440 | 14400 |

We conduct experiments on ETTh1, ETTh2, ETTm1, ETTm2, Weather, Solar, Wind, and ZafNoo (given in (Zhou et al., 2021; Wu et al., 2021; Lai et al., 2018; Li et al., 2022; Poyatos et al., 2020)), which are widely adopted benchmarks in time series research. Notably, except for TimesFM, which has been pretrained on the Weather dataset, no foundation models assert pretraining on these datasets. We have deliberately excluded the Electricity [1] and Traffic (Wu et al., 2021) datasets because numerous TSFMs, including Chronos, TimesFM, TimeMoE, ToTo for Electricity, and Moirai, TimesFM, TimeMoE for Traffic, have been pretrained on them. In addition, we include a self-collected dataset, Service, comprising 20 microservice monitoring metrics from a commercial system, characterized by high sampling rates, long-range physical-time dependencies, and pronounced high-frequency dynamics. The Service dataset is available as part of our open-source replication package.

We evaluate across four forecasting horizons: $L = 96, 192, 336, 720$ for standard LTSF, and $L = 1440$ mainly for Ultra LTSF. Additionally, we perform experiments with longer prediction lengths on some datasets and TSFMs for comprehensive analysis. For the Service dataset, due to its large intrinsic periodic cycle length denoted by $P$, we employ a look-back length of $H = 10P$. For other datasets, we use a look-back length of $H = 30P$ for both baseline TSFMs and SPRINT. In cases where the baseline TSFMs have a model-supported look-back length shorter than $H = 30P$, the look-back length is truncated to the maximum allowable by the model. Conversely, for SPRINT, which can enhance the model-supported look-back length of the base TSFM, a default of $H = 30P$ is maintained. More details of datasets are summarized in Tbl. 6.

## B.3. Metrics

Consistent with prior work (Wu et al., 2023; Shi et al., 2024; Liu et al., 2024), we report Mean Squared Error (MSE) as the evaluation metrics.

## B.4. Implementation Details

The implementation of SPRINT is based on the Time Series Library (Wu et al., 2023), with model and experimental code provided in the supplementary materials. All experiments were conducted using PyTorch (Paszke et al., 2019) on a single NVIDIA L20 GPU equipped with 48GB of memory. In instances where an out-of-memory error occurred, multiple L20 GPUs were utilized to ensure adequate computational resources. For SPRINT, RevIN (Kim et al., 2021) normalization was utilized, alongside a downsampling interval of $k = P/4$, where $P$ represents the intrinsic periodic length of each dataset.

---

[1] https://archive.ics.uci.edu/ml/datasets/ElectricityLoadDiagrams20112014

The Pattern Replication mechanism incorporates a seasonal decay factor $w$ set to 0.9. All experiments were repeated using multiple random seeds, and the mean values of the results were reported. The initial batch size was set at 32, but in cases of an Out-Of-Memory (OOM) error, the batch size could be halved, with a minimum size of 4.

## C. Experimental Results of Ultra LTSF

*Table 7.* Results of Ultra LTSF, which are based on prediction length of $L = 1440$. **Bold** indicates best result. Underlined indicates worse than Seasonal Naive. A slash (/) denotes datasets included in the model's pretraining and thus excluded from testing. A dash (-) indicates that NaN values were present in Timer's raw predictions.

| Dataset | Seasonal Naive | Encoder-Decoder | | Encoder-only | | Decoder-only | | | | | | | | VLM-based | |
|---|---|---|---|---|---|---|---|---|---|---|---|---|---|---|---|
| | | Chronos | +SPRINT | Moirai | +SPRINT | TimesFM | +SPRINT | TimeMoE | +SPRINT | Timer | +SPRINT | ToTo | +SPRINT | VisionTS | +SPRINT |
| ETTh1 | 0.715 | 0.603 | **0.595** | 0.604 | **0.543** | 0.627 | **0.611** | 0.595 | **0.571** | 0.539 | **0.509** | 0.685 | **0.525** | 0.863 | **0.601** |
| ETTh2 | 0.583 | **0.500** | 0.504 | 0.497 | **0.474** | 0.554 | **0.523** | 0.476 | **0.459** | 0.453 | **0.460** | 0.466 | **0.460** | 0.540 | **0.511** |
| ETTm1 | 0.650 | 0.766 | **0.470** | 0.538 | **0.417** | 0.589 | **0.436** | 0.583 | **0.420** | - | 0.401 | 0.531 | **0.414** | 0.731 | **0.490** |
| ETTm2 | 0.531 | 0.493 | **0.384** | 0.478 | **0.376** | 0.503 | **0.383** | 0.423 | **0.355** | - | 0.356 | 0.418 | **0.356** | 0.402 | **0.385** |
| Weather | 0.486 | 0.430 | **0.364** | 0.491 | **0.361** | / | / | 0.665 | **0.339** | - | 0.336 | 0.350 | **0.332** | 0.402 | **0.382** |
| Solar | 0.376 | 0.974 | **0.260** | 0.239 | **0.230** | 0.645 | **0.211** | 0.928 | **0.204** | - | 0.208 | 0.284 | **0.212** | 0.846 | **0.360** |
| Wind | 2.164 | 1.706 | **1.341** | 1.624 | **1.285** | 1.939 | **1.248** | 1.306 | **1.193** | - | 1.189 | 1.517 | **1.232** | 1.267 | **1.237** |
| ZafNoo | 1.088 | 0.937 | **0.777** | 0.844 | **0.796** | 0.857 | **0.788** | 0.768 | **0.713** | - | 0.740 | 0.771 | **0.749** | 1.064 | **0.785** |
| Service | 0.265 | 2.072 | **0.165** | 0.471 | **0.168** | 1.687 | **0.166** | 1.484 | **0.168** | - | - | 0.380 | **0.171** | 0.507 | **0.309** |
| **Avg. Improv.** | - | - | **31.16%** | - | **19.97%** | - | **32.39%** | - | **31.46%** | - | - | - | **18.45%** | - | **22.55%** |

Tbl. 7 presents the results for the challenging ultra LTSF task with a prediction horizon of $L = 1440$. Compared to the standard LTSF results, the performance advantage of SPRINT is even more pronounced in this regime. Quantitatively, SPRINT yields average performance boosts ranging from 18.45% to 32.39% in MSE compared to base TSFMs across the evaluated TSFMs. On high-frequency datasets with strong periodicity like Solar and Service, SPRINT achieves remarkable error reductions, validating that SPRINT effectively mitigates the cumulative error typically associated with extremely long horizons. Furthermore, SPRINT successfully extends the effective prediction lengths of foundation models. Notably, Timer without SPRINT fails to produce valid ultra-long predictions on some datasets. However, equipped with SPRINT, it can seamlessly perform ultra long-term forecasting. This confirms that SPRINT serves as a crucial enabler for scaling existing TSFMs to ultra-long forecasting scenarios.

## D. Analysis of Interpolation Methods

*Table 8.* Comparison of interpolation methods.

| Dataset | Chronos | | | Moirai | | | TimeMoE | | |
|---|---|---|---|---|---|---|---|---|---|
| | Spline | Linear | Nearest | Spline | Linear | Nearest | Spline | Linear | Nearest |
| ETTh2 | 0.3630 | **0.3625** | 0.3640 | **0.3536** | 0.3538 | 0.3549 | **0.3382** | 0.3386 | 0.3399 |
| ETTm2 | 0.2600 | **0.2597** | 0.2615 | **0.2637** | 0.2638 | 0.2652 | **0.2460** | 0.2463 | 0.2478 |
| Weather | 0.2394 | **0.2388** | 0.2408 | 0.2377 | **0.2375** | 0.2388 | 0.2205 | **0.2203** | 0.2222 |
| Service | **0.1536** | 0.1575 | 0.1830 | **0.1614** | 0.1671 | 0.1914 | **0.1529** | 0.1582 | 0.1836 |

In this section, we compare various interpolation methods for trend reconstruction in the Trend Resolution Interpolation module. Consistent with ablation studies (Sec. 5.3), experiments are conducted on 4 datasets with different sampling rates (ETTh2, ETTm2, Weather, and Service) using 3 representative TSFMs: Chronos (encoder-decoder), Moirai (encoder-only), and TimeMoE (decoder-only). As shown in Tbl. 8, both continuous interpolation methods, spline and linear, achieve comparable performance, with spline showing more stable results across different datasets. In contrast, nearest neighbor interpolation performs slightly worse due to its limitation in reconstructing a smooth trend. Overall, SPRINT adopts spline interpolation by default.

## E. Additional Analysis of Enhanced Accuracy Regarding Decomposition

To evaluate whether the accuracy improvements are exclusively attributable to decomposition, we compare SPRINT with a variant named **Season TSFM**. This variant decomposes the time series, handles the trend in the same way as SPRINT, but forces the base TSFM to forecast the high-frequency seasonal component instead of using Pattern Replication.

As shown in Tbl. 9, Season TSFM improves over the base TSFM but remains worse than SPRINT. This indicates that the improvement comes from *both decomposition and the Pattern Replication mechanism*, rather than decomposition alone.

*Table 9.* Comparison with the Season TSFM variant. A slash (/) denotes datasets included in the model's pretraining and thus excluded from testing.

| Model | Variant | ETTh2 | ETTm2 | Weather | Service |
|-------|---------|-------|-------|---------|---------|
| Moirai | Base | 0.372 | 0.286 | 0.254 | 0.304 |
| | +SPRINT | **0.354** | **0.264** | **0.238** | **0.161** |
| | Season TSFM | 0.356 | 0.267 | 0.247 | 0.424 |
| TimesFM | Base | 0.402 | 0.332 | / | 0.954 |
| | +SPRINT | 0.371 | **0.278** | / | **0.157** |
| | Season TSFM | **0.363** | 0.290 | / | 0.588 |
| Timer | Base | 0.356 | 0.279 | 0.243 | 0.745 |
| | +SPRINT | **0.343** | **0.254** | **0.227** | **0.494** |
| | Season TSFM | 0.345 | 0.260 | 0.231 | 0.644 |
| VisionTS | Base | 0.397 | 0.320 | 0.290 | **0.426** |
| | +SPRINT | **0.367** | **0.284** | **0.266** | 0.491 |
| | Season TSFM | 0.399 | 0.315 | 0.292 | 0.962 |

## F. Parameter Sensitivity

We evaluate the robustness of SPRINT by analyzing its sensitivity to three key hyperparameters: the downsampling rate $k$, the seasonal decay factor $w$, and the historical look-back length $H$. All sensitivity experiments are conducted using the Moirai backbone on three representative datasets (ETTh2, ETTm2, Weather) with different sampling rates. Specifically, we investigate: (1) **downsampling interval** $k$: varying the interval $k$ relative to the cycle length $P$ among $\{P/2, P/4, P/8, P/16\}$; (2) **seasonal decay factor** $w$: varying the base weight for the weighted average mechanism among $\{0.7, 0.75, 0.8, 0.85, 0.9, 1\}$; and (3) **historical length** $H$: assessing performance across increasing look-back windows in terms of multiples of $P$ (from $3P$ up to $90P$).

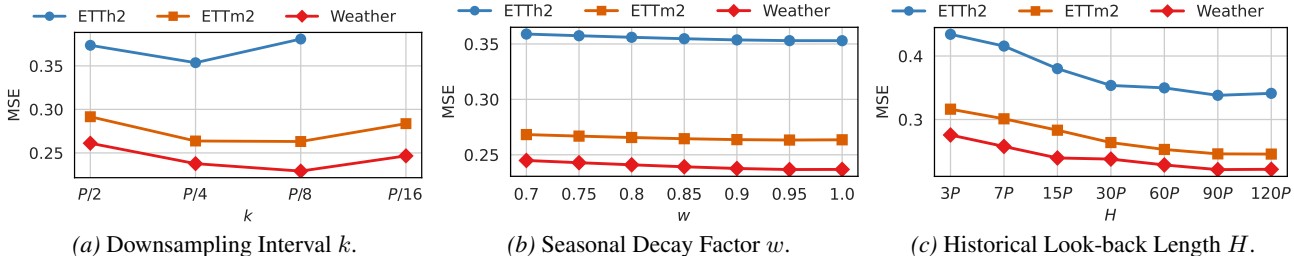

*(a) Downsampling Interval $k$.*      *(b) Seasonal Decay Factor $w$.*      *(c) Historical Look-back Length $H$.*

*Figure 8.* Results of parameter sensitivity analysis. We employ Moirai as backbones on ETTh2, ETTm2 and Weather. Note that cycle length $P = 24$ of ETTh2 does not support $k = P/16$.

**Downsampling interval $k$.** Fig. 8a illustrates the forecasting performance under different downsampling rates. We observe a general trade-off: moderate downsampling (typically around $P/4$ to $P/8$) yields the optimal performance. When the downsampling interval is too large (*e.g.*, $k = P/2$), the sequence becomes too sparse, leading to significant information loss where the interpolated trend fails to capture essential dynamics. Conversely, when the interval is too small (*e.g.*, $k = P/16$), the trend component may still retain excessive high-frequency noise, which can distract the model from focusing on the underlying long-term trajectory.

**Seasonal decay factor $w$.** Fig. 8b analyzes the effect of the decay factor $w$ in the seasonal pattern replication. A value of $w = 1.0$ implies a uniform average of all historical cycles, while smaller values assign higher importance to more recent cycles. On the three datasets, performance tends to degrade as $w$ decreases. When $w$ is greater than 0.9, the performance is almost the same. In practice, considering the influence of recent cycles and potential long-term distribution shifts, we set $w$ to 0.9.

**Historical length $H$.** The impact of the historical look-back length $H$ is shown in fig. 8c. Standard TSFMs often struggle to benefit from extended history due to optimization difficulties or context limitations. In contrast, SPRINT effectively leverages longer historical contexts. We observe that forecasting error generally decreases as $H$ increases util it arrives $90P$, as the downsampled trend component can capture longer-range dependencies without overwhelming the model with

computational complexity. This scalability validates our core motivation that processing in a downsampled space enables the effective utilization of extended historical information for better long-term forecasting. In practice, we recommend setting $H$ between $30P$ and $60P$ to balance performance and efficiency.

## G. Effectiveness of Probabilistic Forecasting

SPRINT seamlessly supports probabilistic forecasting. Specifically, if the base TSFM outputs a predictive distribution (*e.g.*, sampled trajectories), our resolution interpolation workflow is simply applied independently to each predicted trajectory to restore the high-resolution probabilistic forecast.

*Table 10.* Results of probabilistic forecasting. CRPS is reported. A slash (/) denotes datasets included in the model's pretraining and thus excluded from testing.

| Model | Variant | ETTh2 | ETTm2 | Weather | Service |
|---|---|---|---|---|---|
| Chronos | Base | 0.369 | 0.320 | 0.257 | 0.584 |
| | +SPRINT | **0.351** | **0.282** | **0.236** | **0.265** |
| TimesFM | Base | 0.295 | 0.246 | / | 0.457 |
| | +SPRINT | **0.272** | **0.227** | / | **0.277** |

We evaluate probabilistic forecasting using the widely-used CRPS (continuous ranked probability score), where lower values are better. As shown in Tbl. 10, SPRINT-enhanced TSFMs consistently outperform their bases, confirming its robust predictive superiority.

## H. Limitation and future work

While SPRINT exhibits strong performance in enhancing TSFMs, there exist some limitations and directions for future research. First, the Pattern Replication mechanism is most effective for data with identifiable periodic patterns. Although we mitigate issues in non-seasonal series through adaptive downsampling (Sec. 7), future work could explore richer automatic configuration or adaptive gating mechanisms to switch strategies based on the intrinsic data properties. Second, consistent with the mainstream TSFM paradigm, SPRINT adopts a channel-independent strategy, processing multivariate time series individually. While this ensures broad applicability, it may overlook potential correlations between variables. Future research could examine methods to adapt SPRINT to retain these dependencies, particularly during downsampling. Third, our current evaluation focuses on the zero-shot generalization capabilities of TSFMs. However, the potential of SPRINT in fine-tuning scenarios remains unexplored. Future work could investigate introducing learnable components (*e.g.*, learnable representative patterns) within SPRINT or joint fine-tuning of TSFMs to further enhance performance when some training data is available.

## I. Full Results

## J. Showcases

### J.1. Showcases of overall forecasting

We present the visualization of overall forecasting results for different TSFMs across three datasets: ETTh2 ($L = 96 = 4P$), ETTm2 ($L = 336 = 3.5P$), and Weather ($L = 720 = 5P$), in Fig. 9, 10, 11, 12, 13, 14, 15.

### J.2. Showcases of seasonal prediction

To highlight the efficacy of the Pattern Replication mechanism in capturing periodicity, we visualize the season predictions. Fig. 16 demonstrates season predictions across 9 diverse datasets with their respective prediction lengths. Additionally, Fig. 17 examines the robustness of the mechanism on the ETTh1 dataset under varying prediction lengths $L$, ranging from 96 to 720.

*Table 11.* Detailed results of LTSF with prediction lengths of $L \in \{96, 192, 336, 720\}$.

| Methods | | Chronos | +Ours | Moirai | +Ours | TimesFM | +Ours | TimeMoE | +Ours | Timer | +Ours | ToTo | +Ours | VisionTS | +Ours |
|---|---|---|---|---|---|---|---|---|---|---|---|---|---|---|---|
| ETTh1 | 96 | 0.484 | **0.433** | 0.376 | **0.371** | 0.421 | **0.375** | **0.349** | 0.362 | 0.370 | **0.363** | 0.377 | **0.357** | 0.775 | **0.468** |
| | 192 | 0.538 | **0.468** | 0.426 | **0.409** | 0.472 | **0.413** | **0.398** | 0.398 | 0.406 | **0.395** | 0.424 | **0.394** | 0.767 | **0.489** |
| | 336 | 0.554 | **0.474** | 0.471 | **0.421** | 0.510 | **0.438** | 0.426 | **0.420** | 0.433 | **0.403** | 0.465 | **0.414** | 0.736 | **0.495** |
| | 720 | 0.530 | **0.486** | 0.509 | **0.440** | 0.514 | **0.464** | 0.463 | **0.443** | 0.446 | **0.415** | 0.498 | **0.417** | 0.834 | **0.516** |
| ETTh2 | 96 | 0.307 | **0.293** | 0.295 | **0.294** | 0.326 | **0.315** | 0.281 | **0.279** | 0.291 | **0.288** | 0.289 | **0.267** | 0.348 | **0.308** |
| | 192 | 0.377 | **0.361** | 0.365 | **0.356** | 0.399 | **0.373** | 0.335 | **0.332** | 0.353 | **0.343** | 0.365 | **0.331** | 0.395 | **0.368** |
| | 336 | 0.415 | **0.386** | 0.403 | **0.376** | 0.433 | **0.383** | 0.357 | **0.353** | 0.382 | **0.356** | 0.405 | **0.353** | 0.404 | **0.386** |
| | 720 | 0.432 | **0.412** | 0.427 | **0.389** | 0.450 | **0.411** | 0.393 | **0.389** | 0.400 | **0.386** | 0.421 | **0.373** | 0.439 | **0.407** |
| ETTm1 | 96 | 0.510 | **0.371** | **0.285** | 0.319 | 0.357 | **0.322** | **0.268** | 0.308 | **0.307** | 0.326 | **0.310** | 0.311 | 0.687 | **0.422** |
| | 192 | 0.599 | **0.401** | 0.343 | **0.342** | 0.412 | **0.343** | 0.347 | **0.332** | 0.356 | **0.346** | 0.361 | **0.335** | 0.689 | **0.440** |
| | 336 | 0.648 | **0.423** | 0.383 | **0.362** | 0.441 | **0.366** | 0.410 | **0.353** | 0.410 | **0.363** | 0.391 | **0.354** | 0.691 | **0.462** |
| | 720 | 0.705 | **0.460** | 0.465 | **0.402** | 0.507 | **0.406** | 0.488 | **0.392** | 0.509 | **0.391** | 0.457 | **0.391** | 0.768 | **0.484** |
| ETTm2 | 96 | 0.213 | **0.180** | **0.174** | 0.183 | 0.205 | **0.190** | 0.180 | **0.168** | 0.191 | **0.175** | 0.169 | **0.167** | 0.256 | **0.213** |
| | 192 | 0.283 | **0.230** | 0.244 | **0.236** | 0.293 | **0.249** | 0.255 | **0.219** | 0.244 | **0.225** | 0.230 | **0.219** | 0.296 | **0.259** |
| | 336 | 0.345 | **0.278** | 0.310 | **0.284** | 0.365 | **0.304** | 0.322 | **0.268** | 0.295 | **0.274** | 0.287 | **0.265** | 0.334 | **0.300** |
| | 720 | 0.444 | **0.352** | 0.417 | **0.352** | 0.464 | **0.371** | 0.387 | **0.330** | 0.386 | **0.340** | 0.367 | **0.334** | 0.394 | **0.365** |
| Weather | 96 | 0.207 | **0.169** | **0.158** | 0.166 | / | / | 0.264 | **0.153** | 0.169 | **0.165** | **0.143** | 0.155 | 0.222 | **0.194** |
| | 192 | 0.257 | **0.211** | **0.206** | 0.208 | / | / | 0.318 | **0.194** | 0.212 | **0.203** | **0.187** | 0.192 | 0.263 | **0.240** |
| | 336 | 0.310 | **0.258** | 0.271 | **0.257** | / | / | 0.407 | **0.238** | 0.261 | **0.243** | 0.238 | **0.234** | 0.308 | **0.289** |
| | 720 | 0.379 | **0.319** | 0.379 | **0.320** | / | / | 0.542 | **0.297** | 0.330 | **0.297** | 0.304 | **0.289** | 0.366 | **0.343** |
| Solar | 96 | 0.774 | **0.236** | **0.193** | 0.218 | 0.363 | **0.204** | 0.267 | **0.191** | 0.351 | **0.207** | 0.231 | **0.208** | 0.684 | **0.332** |
| | 192 | 0.859 | **0.253** | **0.216** | 0.227 | 0.410 | **0.209** | 0.427 | **0.202** | 0.473 | **0.210** | 0.261 | **0.215** | 0.717 | **0.346** |
| | 336 | 0.916 | **0.264** | **0.231** | 0.236 | 0.475 | **0.217** | 0.596 | **0.213** | 0.579 | **0.216** | 0.278 | **0.220** | 0.706 | **0.351** |
| | 720 | 0.954 | **0.265** | 0.243 | **0.237** | 0.551 | **0.217** | 0.764 | **0.211** | 0.692 | **0.215** | 0.270 | **0.218** | 0.874 | **0.368** |
| Wind | 96 | 1.017 | **0.995** | **0.909** | 0.985 | 1.230 | **0.961** | **0.898** | 0.913 | **0.920** | 0.947 | **0.881** | 0.925 | 1.123 | **1.045** |
| | 192 | 1.264 | **1.142** | **1.098** | 1.136 | 1.504 | **1.065** | 1.074 | **1.041** | 1.082 | **1.072** | 1.099 | **1.056** | 1.190 | **1.133** |
| | 336 | 1.486 | **1.262** | 1.269 | **1.252** | 1.739 | **1.150** | 1.173 | **1.139** | 1.226 | **1.153** | 1.274 | **1.169** | 1.235 | **1.220** |
| | 720 | 1.701 | **1.342** | 1.473 | **1.306** | 1.981 | **1.221** | 1.250 | **1.196** | 1.363 | **1.192** | 1.429 | **1.250** | 1.289 | **1.245** |
| ZafNoo | 96 | 0.515 | **0.461** | **0.447** | 0.461 | 0.517 | **0.447** | 0.436 | **0.429** | **0.437** | 0.438 | **0.428** | 0.433 | 0.842 | **0.468** |
| | 192 | 0.594 | **0.521** | 0.528 | **0.523** | 0.604 | **0.506** | 0.501 | **0.484** | 0.506 | **0.497** | 0.499 | **0.493** | 1.000 | **0.524** |
| | 336 | 0.655 | **0.561** | 0.607 | **0.577** | 0.661 | **0.556** | 0.557 | **0.527** | 0.580 | **0.538** | 0.566 | **0.545** | 1.010 | **0.554** |
| | 720 | 0.769 | **0.640** | 0.752 | **0.659** | 0.749 | **0.640** | 0.640 | **0.591** | 0.712 | **0.607** | 0.654 | **0.626** | 0.906 | **0.622** |
| Service | 96 | 0.413 | **0.148** | 0.221 | **0.157** | 0.364 | **0.151** | 0.375 | **0.146** | **0.296** | 0.317 | **0.131** | 0.167 | **0.366** | 0.536 |
| | 192 | 0.706 | **0.153** | 0.267 | **0.162** | 0.664 | **0.157** | 0.465 | **0.152** | 0.459 | **0.426** | **0.147** | 0.174 | **0.408** | 0.541 |
| | 336 | 1.121 | **0.156** | 0.321 | **0.164** | 1.093 | **0.160** | 0.541 | **0.156** | 0.777 | **0.553** | **0.164** | 0.176 | **0.424** | 0.503 |
| | 720 | 1.785 | **0.157** | 0.406 | **0.162** | 1.697 | **0.160** | 0.919 | **0.158** | 1.447 | **0.680** | 0.222 | **0.169** | 0.506 | **0.385** |

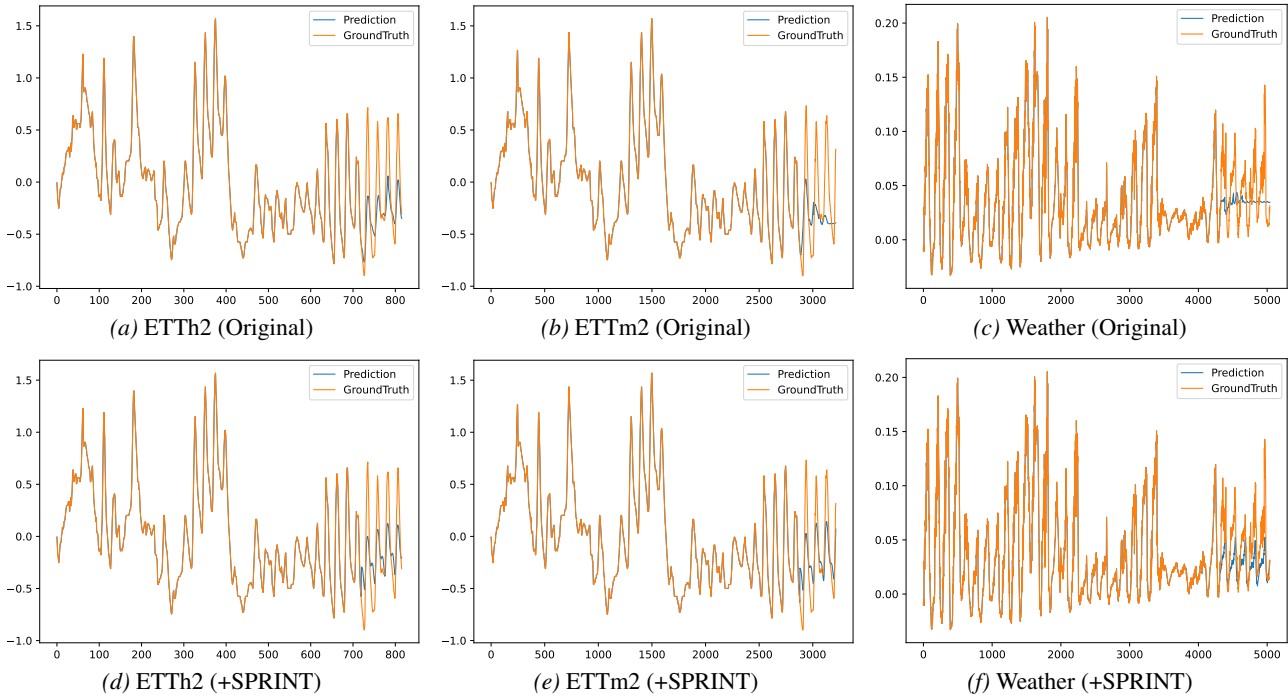

*Figure 9.* Forecasting showcase of **Chronos** with and without SPRINT. The top row displays the predictions from the original TSFM while the bottom row shows the results enhanced by SPRINT. The orange lines represent the ground truth, and the blue lines indicate the model predictions.

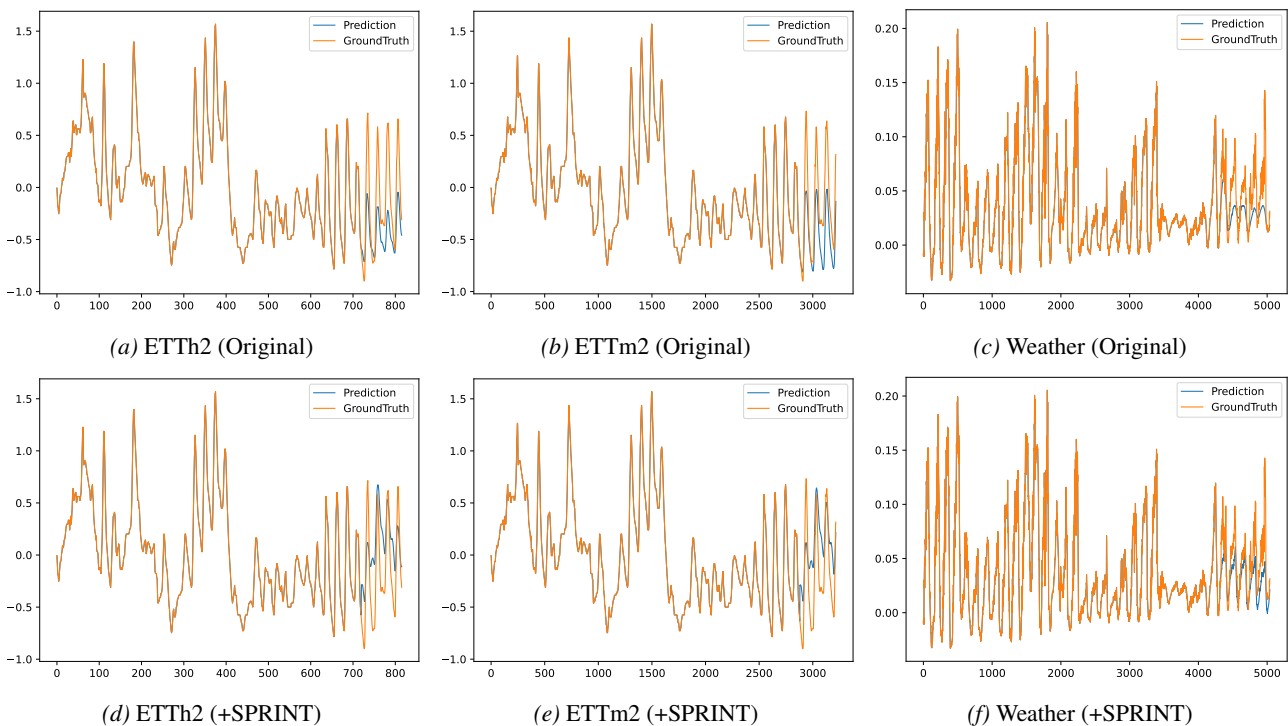

*Figure 10.* Forecasting showcase of **Moirai** with and without SPRINT. The top row displays the predictions from the original TSFM while the bottom row shows the results enhanced by SPRINT. The orange lines represent the ground truth, and the blue lines indicate the model predictions.

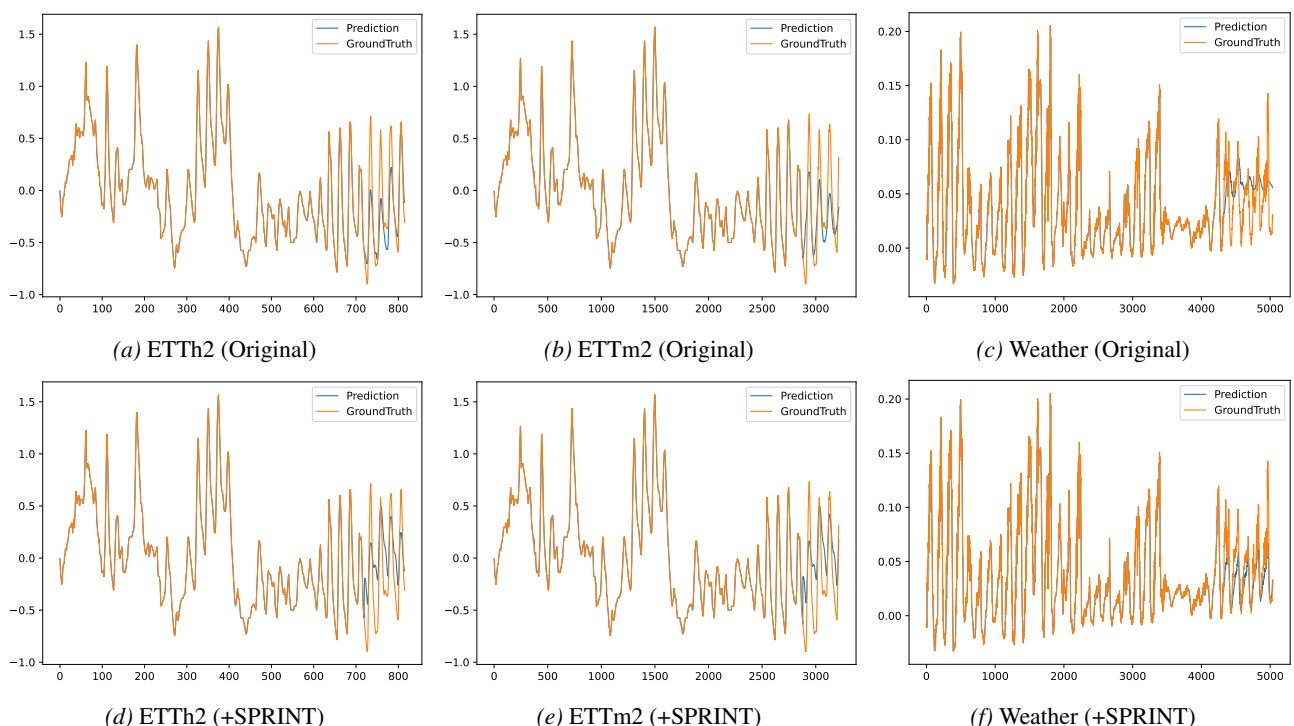

*Figure 11.* Forecasting showcase of **TimesFM** with and without SPRINT. The top row displays the predictions from the original TSFM while the bottom row shows the results enhanced by SPRINT. The orange lines represent the ground truth, and the blue lines indicate the model predictions.

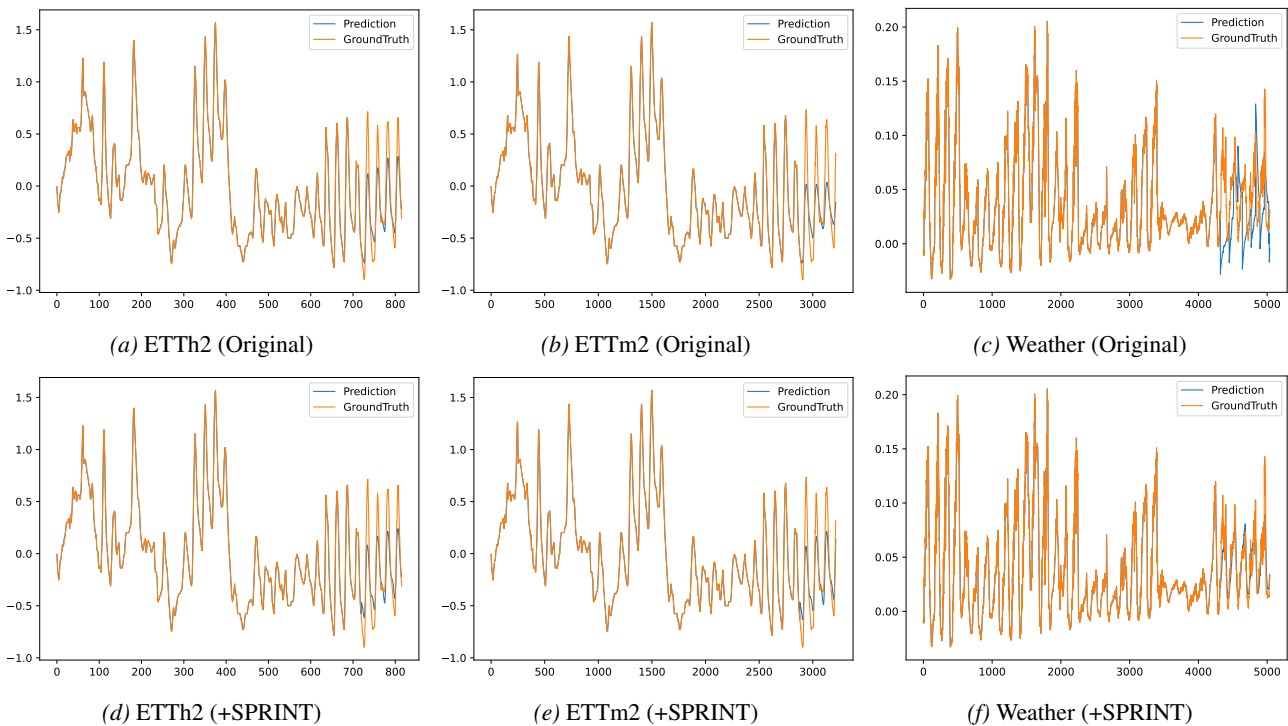

*Figure 12.* Forecasting showcase of **TimeMoE** with and without SPRINT. The top row displays the predictions from the original TSFM while the bottom row shows the results enhanced by SPRINT. The orange lines represent the ground truth, and the blue lines indicate the model predictions.

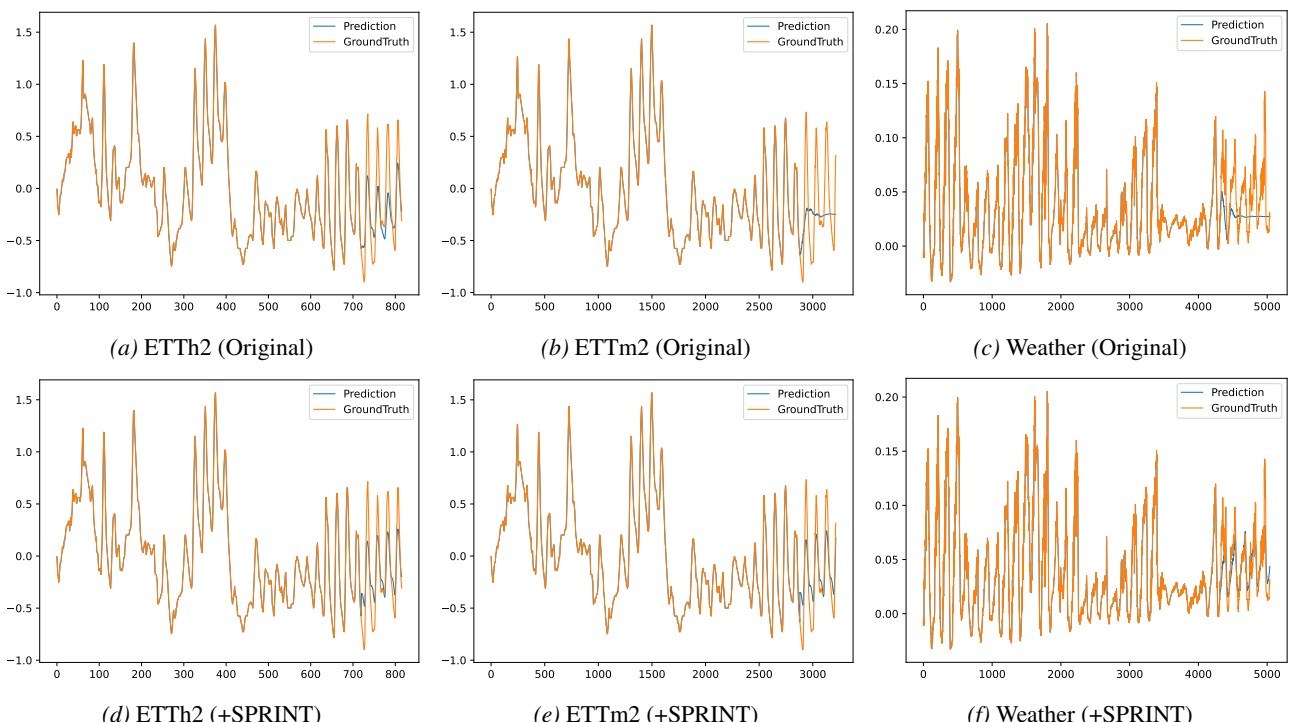

*Figure 13.* Forecasting showcase of **Timer** with and without SPRINT. The top row displays the predictions from the original TSFM while the bottom row shows the results enhanced by SPRINT. The orange lines represent the ground truth, and the blue lines indicate the model predictions.

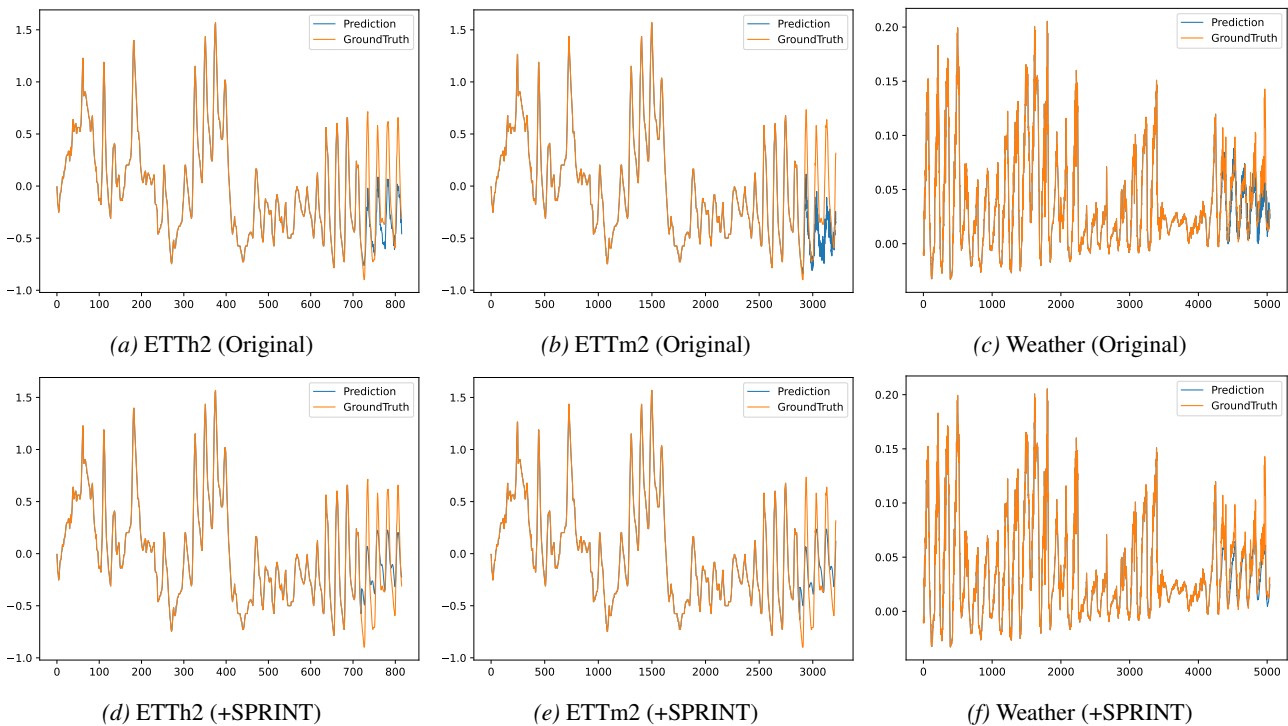

*Figure 14.* Forecasting showcase of **ToTo** with and without SPRINT. The top row displays the predictions from the original TSFM while the bottom row shows the results enhanced by SPRINT. The orange lines represent the ground truth, and the blue lines indicate the model predictions.

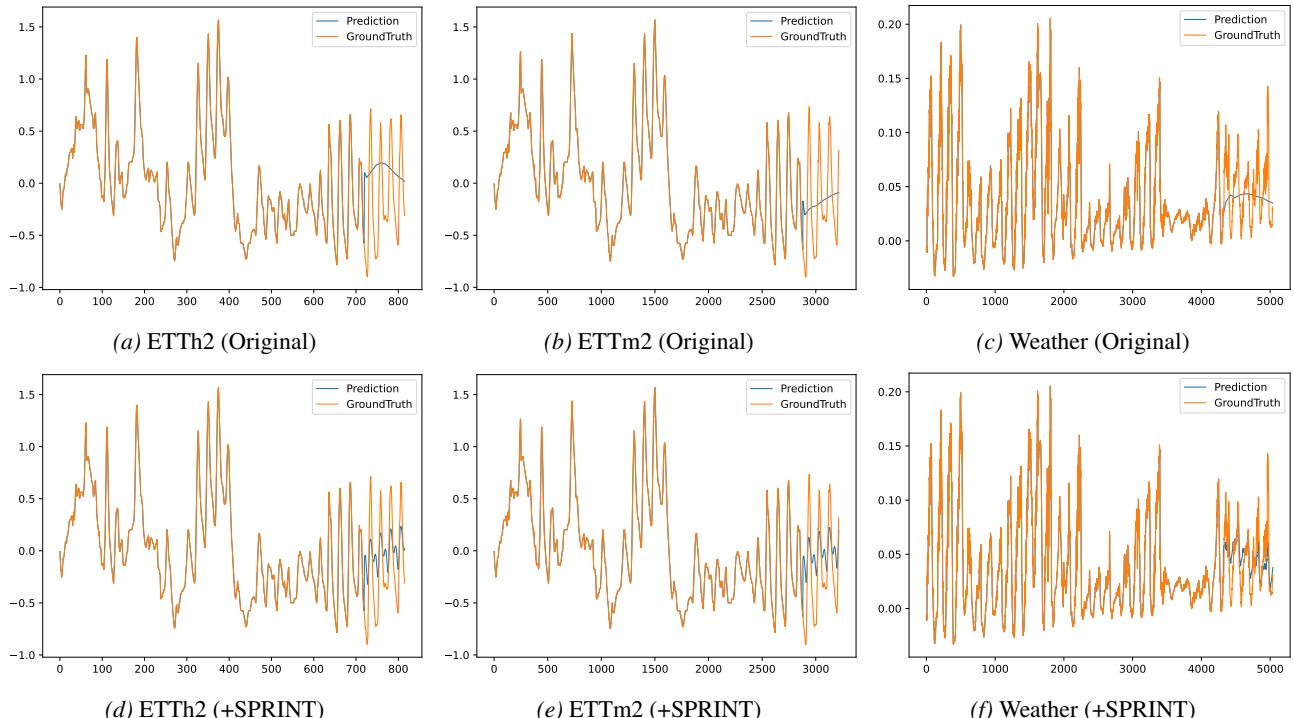

*Figure 15.* Forecasting showcase of **VisionTS** with and without SPRINT. The top row displays the predictions from the original TSFM while the bottom row shows the results enhanced by SPRINT. The orange lines represent the ground truth, and the blue lines indicate the model predictions.

## J.3. Showcases of low-resolution trend prediction

We illustrate the low-resolution trend predictions generated by different TSFMs across the ETTh2, ETTm2, and Weather datasets in Fig. 18, 19, 20, 21, 22, 23, 24.

## J.4. Showcases of trend reconstruction

To evaluate the information loss and recovery capability of our Resolution-Interpolation workflow, we present the reconstruction results of the trend component. Fig. 25 displays the trend reconstruction quality across 9 datasets with a fixed downsampling interval $k = P/4$. Furthermore, to analyze the impact of the downsampling rate, Fig. 26 compares reconstruction results on the ETTm1 dataset under varying $k$ values ($P/2, P/4, P/8, P/16$).

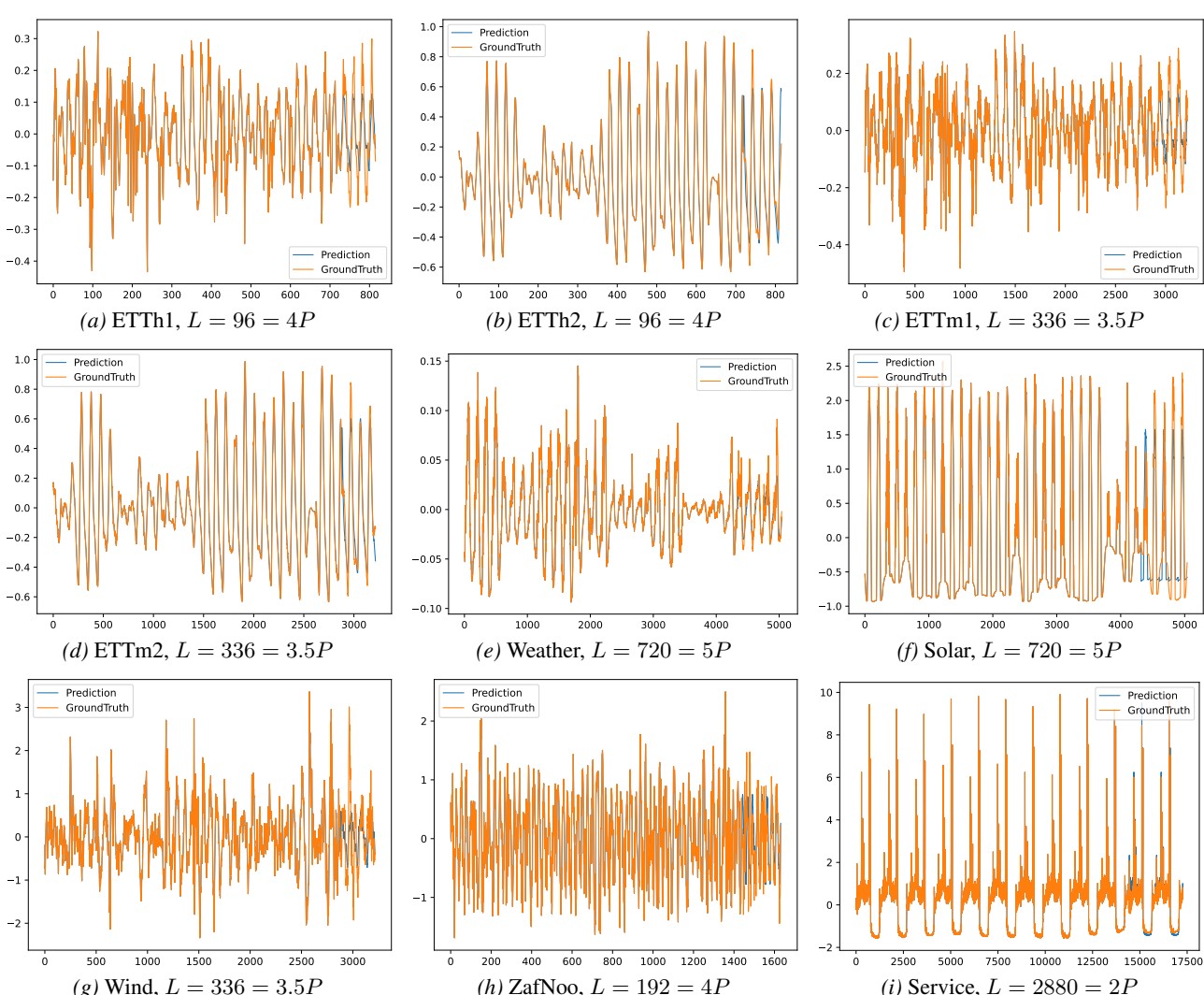

*Figure 16.* Season prediction cases via the Pattern Replication mechanism of SPRINT from different datasets. The look-back length $H$ of each dataset can be found in Appendix B.2. orange lines are the ground truths and blue lines are the interpolation results.

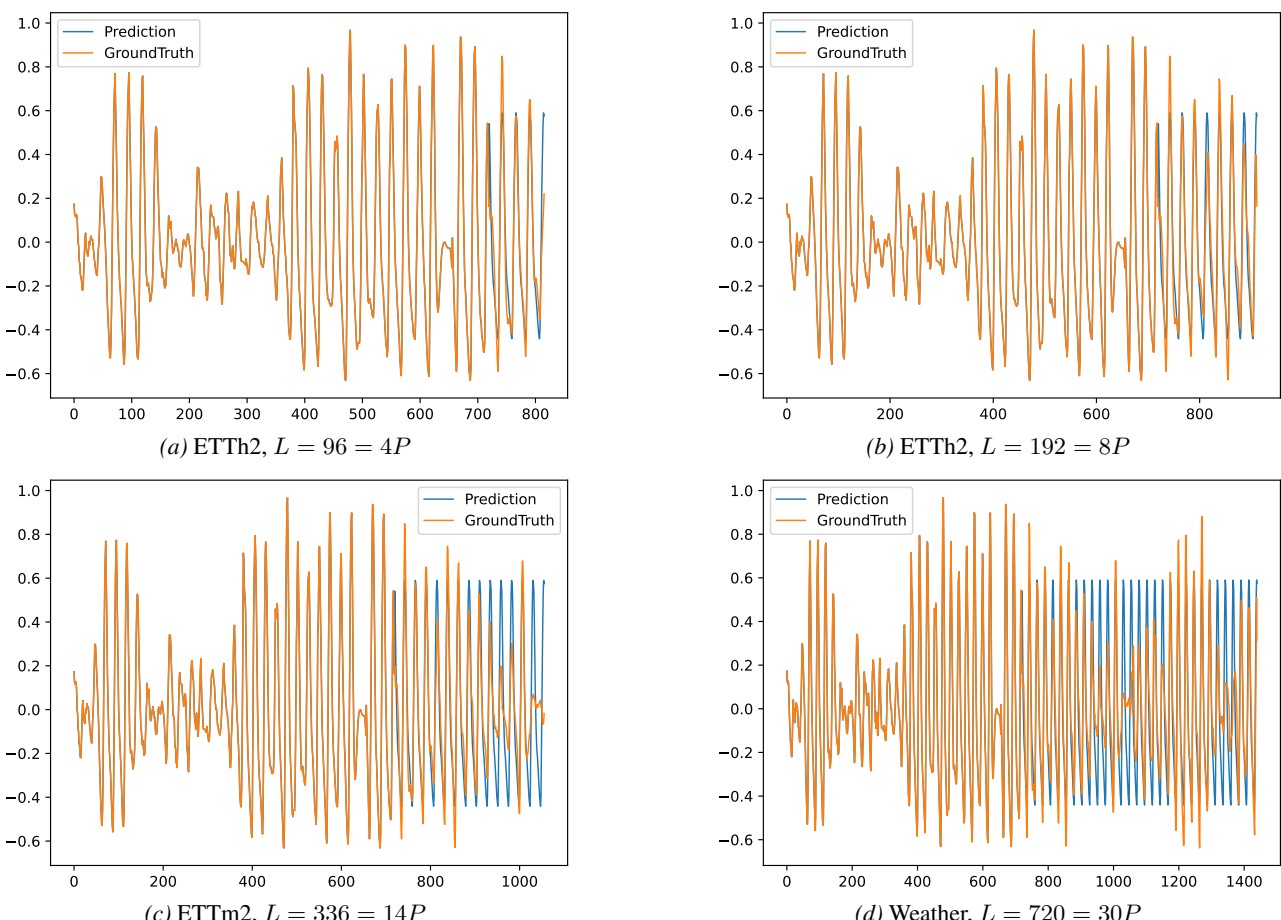

*Figure 17.* Season prediction cases via the Pattern Replication mechanism of SPRINT from ETTh1 with varying prediction lengths $L$. The look-back length $H$ is set to $30P = 720$. orange lines are the ground truths and blue lines are the interpolation results.

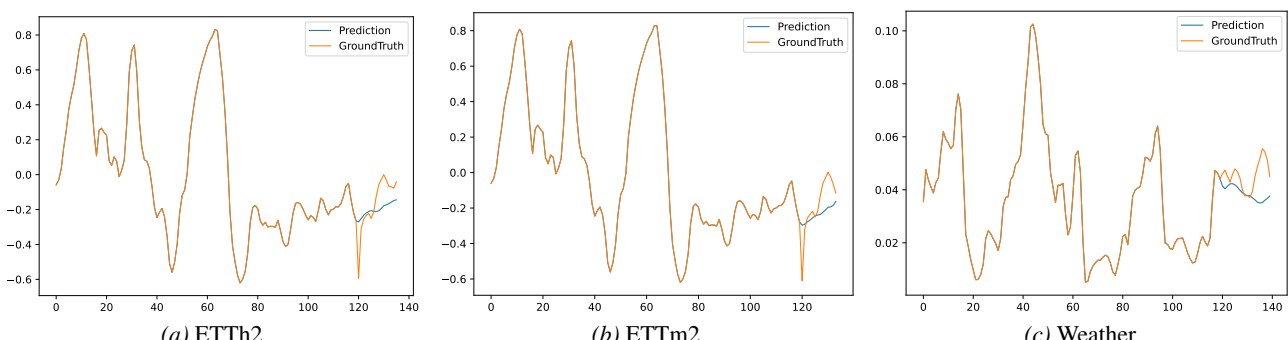

*Figure 18.* Low-resolution trend prediction showcase of **Chronos**.

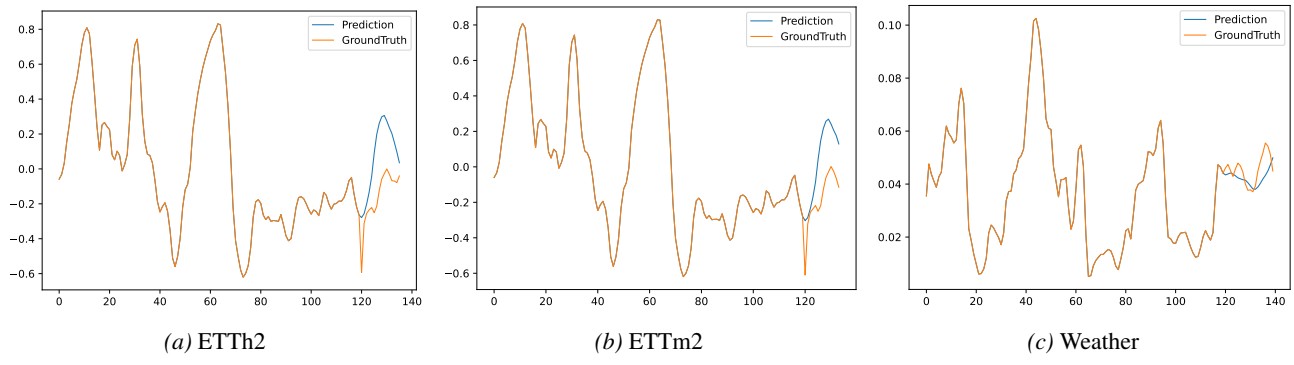

*(a)* ETTh2        *(b)* ETTm2        *(c)* Weather

*Figure 19.* Low-resolution trend prediction showcase of **Moirai**.

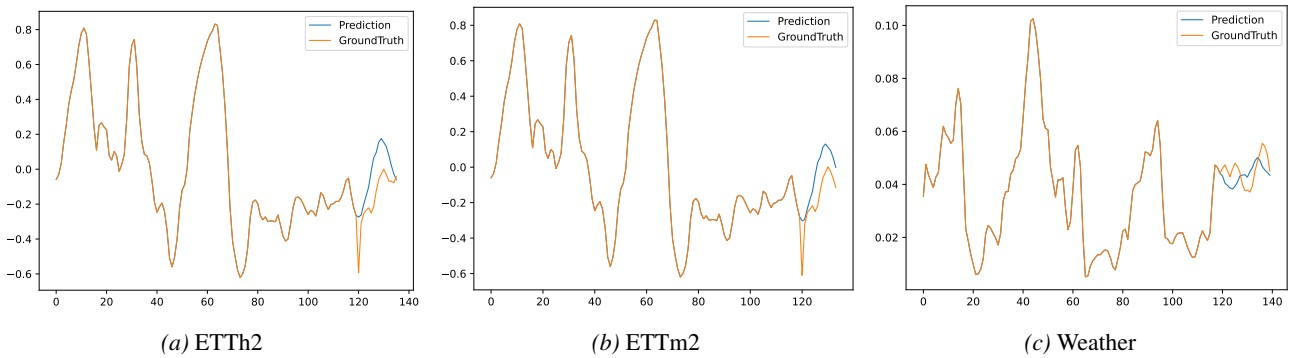

*(a)* ETTh2        *(b)* ETTm2        *(c)* Weather

*Figure 20.* Low-resolution trend prediction showcase of **TimesFM**.

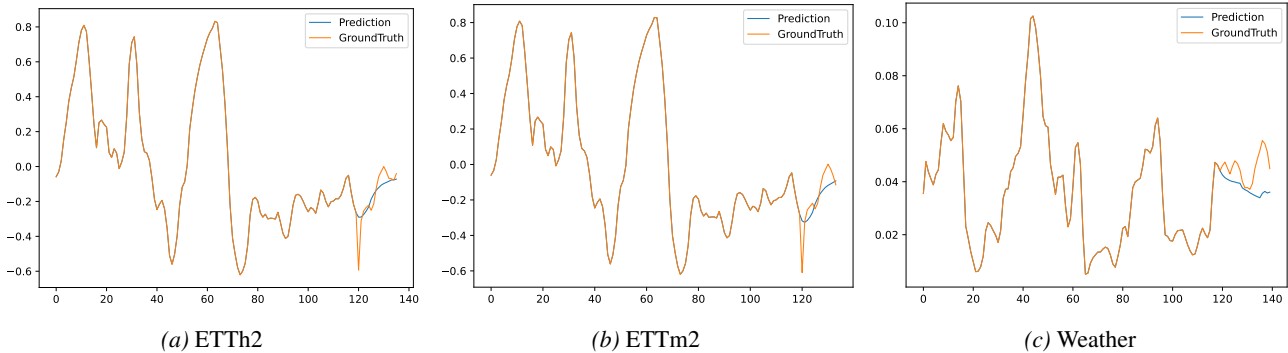

*(a)* ETTh2        *(b)* ETTm2        *(c)* Weather

*Figure 21.* Low-resolution trend prediction showcase of **TimeMoE**.

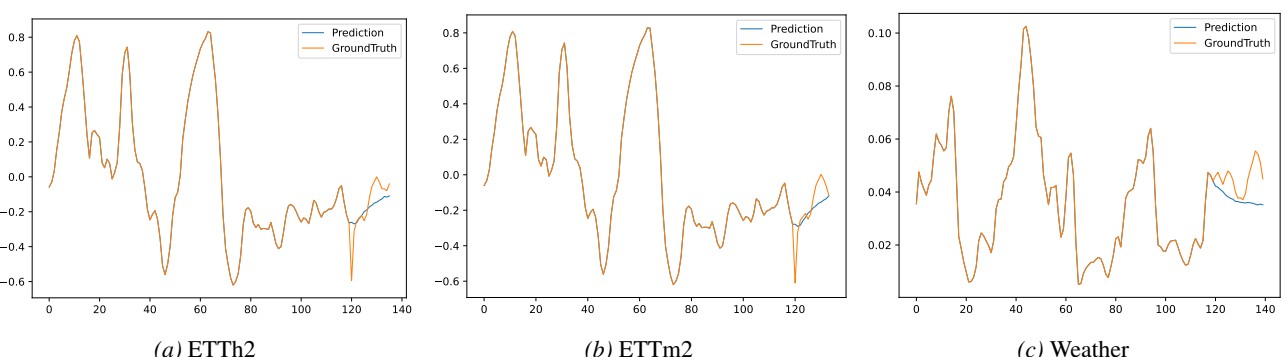

*(a)* ETTh2        *(b)* ETTm2        *(c)* Weather

*Figure 22.* Low-resolution trend prediction showcase of **Timer**.

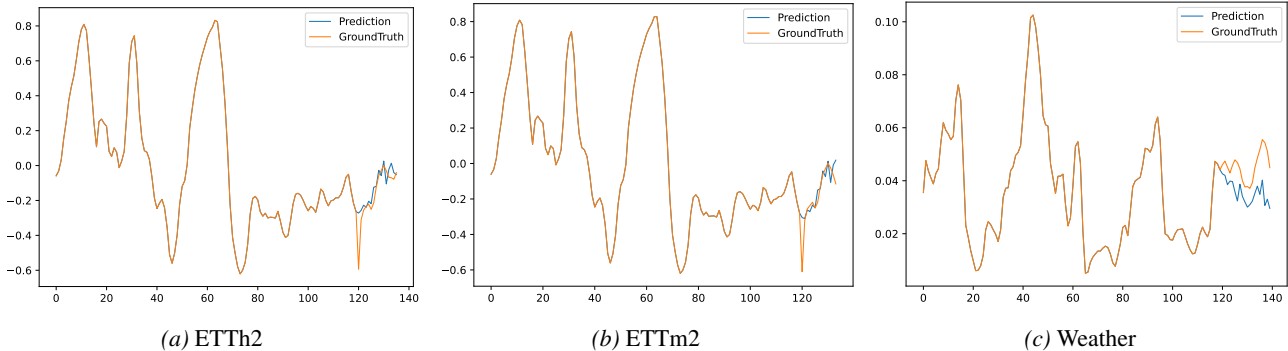

*(a)* ETTh2      *(b)* ETTm2      *(c)* Weather

*Figure 23.* Low-resolution trend prediction showcase of **ToTo**.

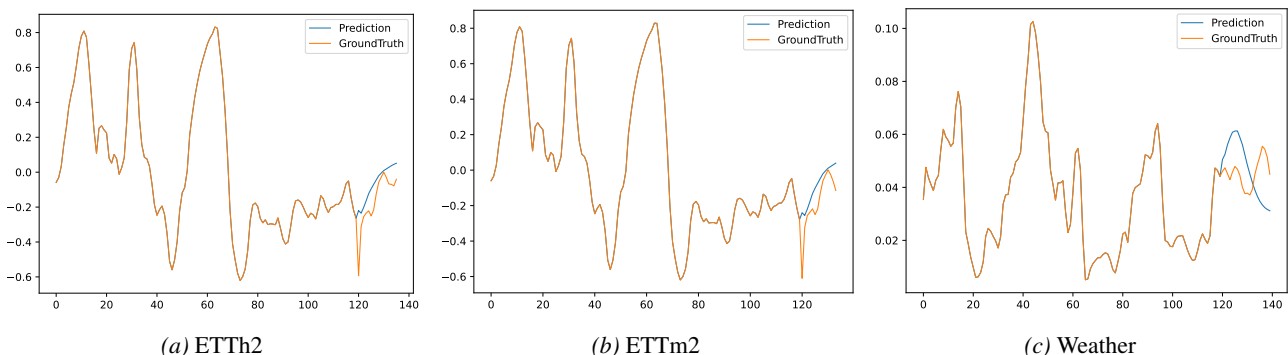

*(a)* ETTh2      *(b)* ETTm2      *(c)* Weather

*Figure 24.* Low-resolution trend prediction showcase of **VisionTS**.

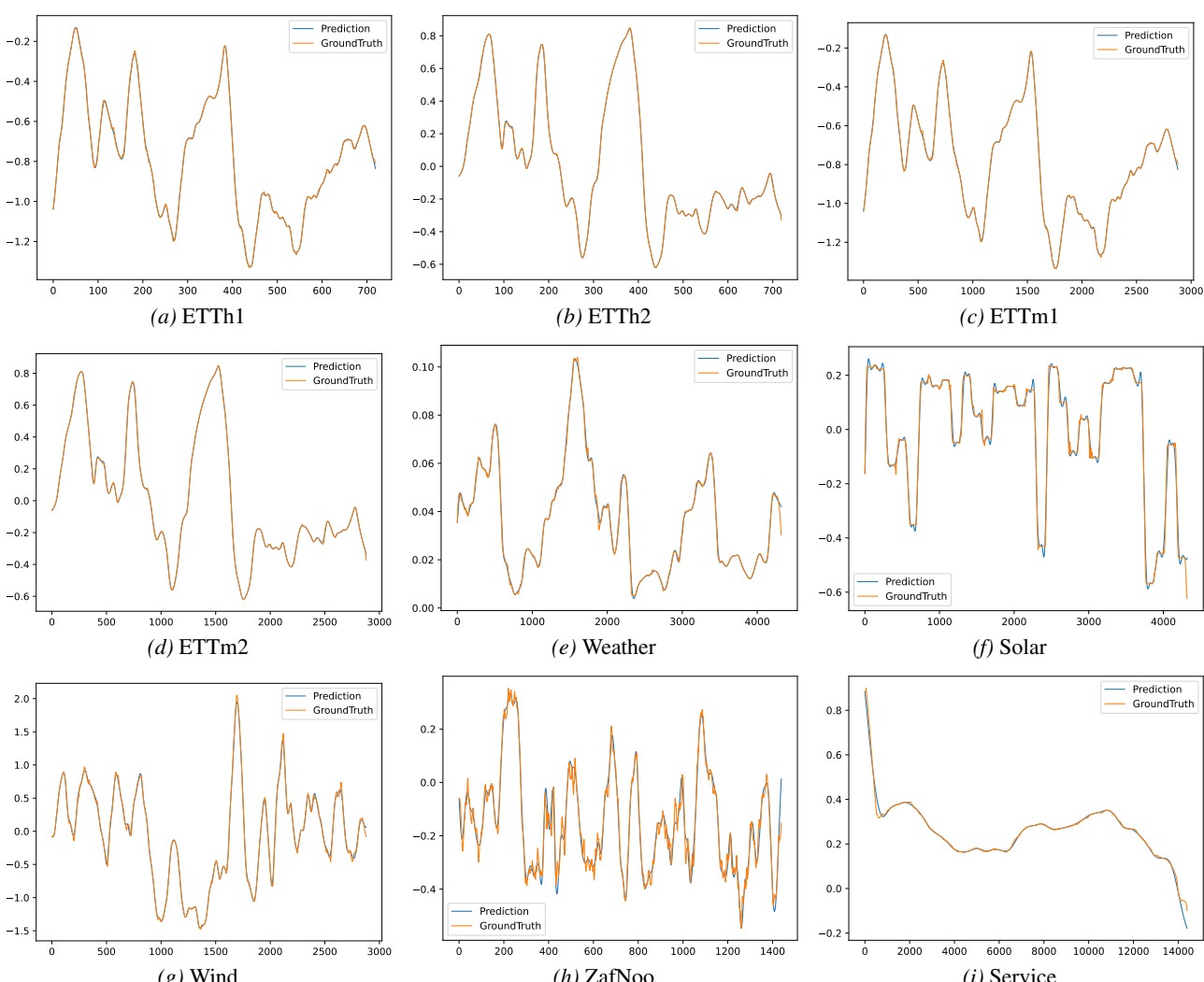

*Figure 25.* Trend interpolation cases from different datasets. Downsampling interval $k$ is set to $P/4$ for each dataset. The original length of series is equal to look-back length $H$ of each dataset (see Appendix B.2). orange lines are the ground truths and blue lines are the interpolation results.

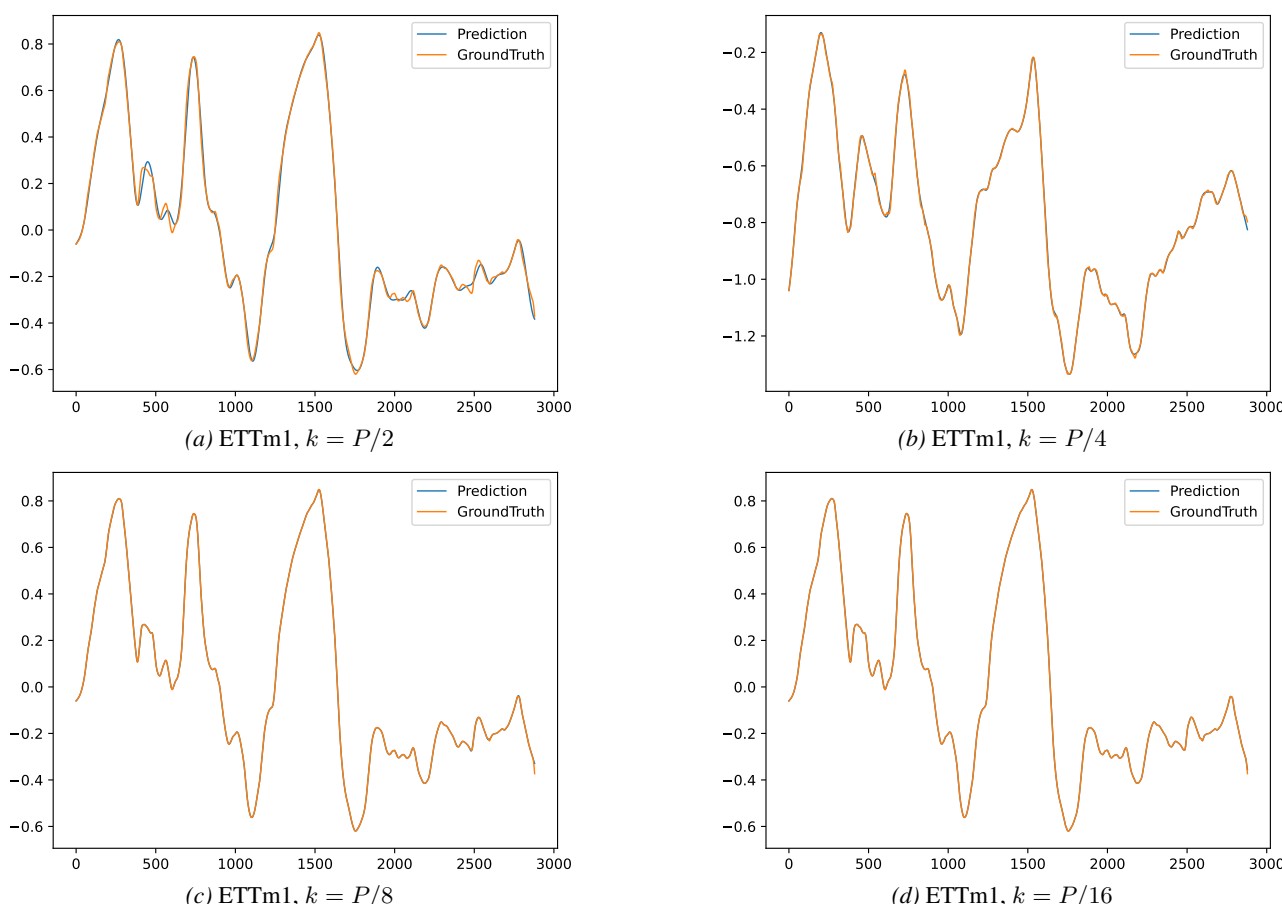

*Figure 26.* Trend interpolation cases from ETTm1 with different downsampling intervals $k$. orange lines are the ground truths and blue lines are the interpolation results.

