# OpenReview forum: "See More, Forecast Better and Faster: Enhancing Time Series Foundation Models via Inference-Time Plug-and-Play Downsampling"
_ICML.cc/2026/Conference — ICML 2026 regular_

### Official Review · Reviewer_kNnK · 2026-03-10

**Soundness:** 4
**Presentation:** 4
**Significance:** 3
**Originality:** 4
**Overall Recommendation:** 4
**Confidence:** 3

**Summary:**

This paper proposes SPRINT, a training-free and plug-and-play framework designed to enhance the long-term forecasting capability of TSFMs.
It decomposes the forecasting task into two components: low-frequency trend forecasting and high-frequency seasonal pattern reconstruction.
This dual-stream approach empowers existing TSFMs to digest longer look-back windows and generate further into the future during inference, completely bypassing the need to fine-tune or retrain the base model.
This framework is well evaluated on multiple benchmark datasets and demonstrates improved performance compared to existing TSFMS.
It boosts accuracy while cutting computational costs.

**Compliance With Llm Reviewing Policy:**

Affirmed.

**Final Justification:**

Thank authors for providing a complete explanation, especially addressing my concerns about high-frequency and seasonal data. I have raised the 'soundness' score to 4.

**Key Questions For Authors:**

Please refer to **Weaknesses**.

**Limitations:**

Yes

**Strengths And Weaknesses:**

**Strengths:**
- SPRINT is a training-free and plug-and-play framework that enhances existing TSFMs at inference time. The rich experiments and forecasting showcases have shown its effectiveness. The experiments test a wide range of foundation model paradigms, including encoder-decoder models, encoder-only models, decoder-only models, and even a VLM-based model.
- This paper does not rely solely on empirical results. It mathematically justifies their design choices, using the Nyquist-Shannon Sampling Theorem to validate trend reconstruction and bounding the cumulative error of pattern replication to polynomial growth.
- It gains a great reduction in computational overhead.


**Weaknesses:**
- SPRINT uses a basic moving average to separate the trend and seasonal components. It may face challenges in highly volatile, non-stationary, or irregular real-world data environments. This framework bypasses the foundation model for forecasting high-frequency data.  It may cause the ignorance of TSFM's learning capacity for complex, high-frequency dynamics.
- The core design hinges on periodic patterns. While this paper offers an 'Adaptive Strategy' for non-periodic data, the strategy relies on manually selecting a downsampling interval, which is determined heuristically.
- The Reference section should be restructured for consistency. Several arXiv papers are cited using inconsistent formats, such as paper 'Visionts: Visual masked autoencoders are free-lunch zero-shot time series forecasters' and 'This time is different: An observability perspective on time series foundation models'.

---

> ### Author Rebuttal · Authors · 2026-03-30
>
> **Thank you for your kind and careful review!**
>
> **Q1: Limitation of moving average.**
>
> **A1:** While the moving average (MA) is simple, it is **widely adopted** for seasonal-trend decomposition in LTSF literature, such as Autoformer [1] and DLinear [2]. It acts as a low-pass filter, retaining the complex, non-stationary distribution shifts and volatility within the trend component. This explicitly *offloads these difficult dynamics to TSFMs*.
>
> Besides, as a plug-and-play framework, SPRINT’s decomposition module is **swappable**. In our ablation study, we replace MA with STL decomposition. But in most cases, STL performs worse than MA and incurs unacceptable computational costs. Thus, MA provides the optimal balance of speed and sufficient decoupling for TSFMs.
>
> **Q2: Limitation of bypassing TSFMs for high-frequency data.**
>
> **A2:** While it may seem intuitive to utilize TSFMs for all components, directly applying TSFMs to the seasonal component can lead to limited improvements in prediction accuracy and result in efficiency issues.
>
> We decompose the series and force the base TSFM to predict the high-frequency seasonal component (denoted as "Season TSFM"), while processing the trend same as SPRINT. The results, compared to both base TSFMs and SPRINT, are shown below, with a slash (/) indicating datasets used in the model’s pretraining.
>
> |Model|Variant|ETTh2|ETTm2|Weather|Service|
> |:-|:-|:-|:-|:-|:-|
> |Moirai|Base|0.372|0.286|0.254|0.304|
> ||+SPRINT|**0.354**|**0.264**|**0.238**|**0.161**|
> ||Season TSFM|0.356|0.267|0.247|0.424|
> |TimesFM|Base|0.402|0.332|/|0.954|
> ||+SPRINT|0.371|**0.278**|/|**0.157**|
> ||Season TSFM|**0.363**|0.290|/|0.588|
> |Timer|Base|0.356|0.279|0.243|0.745|
> ||+SPRINT|**0.343**|**0.254**|**0.227**|**0.494**|
> ||Season TSFM|0.345|0.260|0.231|0.644|
> |VisionTS|Base|0.397|0.320|0.290|**0.426**|
> ||+SPRINT|**0.367**|**0.284**|**0.266**|0.491|
> ||Season TSFM|0.399|0.315|0.292|0.962|
>
> Results show that "Season TSFM" yields higher errors compared to SPRINT. However, it still outperforms the base TSFM. This confirms that the overall effectiveness of SPRINT stems from **BOTH the decomposition strategy and the pattern replication mechanism**.
>
> **Q3: Limitation of heuristically selected downsampling interval ($k$).**
>
> **A3:** To eliminate manual tuning, we extend this adaptive strategy to a **data-driven, automated $k$-selection** mechanism that maps the intrinsic data smoothness to the optimal downsampling interval ($k_{opt}$). We will include this mechanism with experimental analysis in the appendix.
>
> The smoothness of each series, denotes as $s$, is quantified as the product of the **Turning Points Ratio** and **Normalized Volatility**, where a lower value indicates a smoother series. The Turning Points Ratio measures the proportion of local extrema, capturing the frequency of directional changes. The Normalized Volatility is defined as the normalized mean of the absolute differences between adjacent data points, quantifying the step-wise fluctuations.
>
> We conduct an empirical study of the correlation between $k_{opt}$ and $s$, using 3 financial datasets which are day-sampled and non-periodic (look-back length $H=64$, prediction length $L\in\{24,36,48,60\}$). To get different $s$, we apply moving average on raw datasets and get smoothed variants (e.g., `S4` denotes an MA window size of 4).
>
> |Dataset|Variant|Smoothness $s$ (1e-3)$\downarrow$|$k_{opt}$|MSE|
> |:-|:-|:-|:-|:-|
> |Exchange|Raw|10.5|1|0.0395|
> ||S2|5.60|2|0.0379|
> ||S4|2.85|2|0.0386|
> ||S8|1.38|4|0.0396|
> ||S16|0.720|8|0.0396|
> |NASDAQ|Raw|12.3|1|1.040|
> ||S2|7.12|1|1.080|
> ||S4|4.36|4|1.096|
> ||S8|2.62|4|0.955|
> ||S16|1.50|4|0.829|
> |NYSE|Raw|6.15|4|0.637|
> ||S2|3.73|4|0.638|
> ||S4|2.33|4|0.639|
> ||S8|1.40|8|0.644|
> ||S16|0.853|8|0.599|
>
> Results show that smoother data consistently correlates with a larger $k_{opt}$. We plot the smoothness-$k_{opt}$ relationship at https://anonymous.4open.science/r/icml11771-rebuttal-007D/smoothness_vs_downsampling_interval.pdf. Across all datasets, the Pearson correlation coefficient between $\log_2(k_{opt})$ and the smoothness metric is **-0.859**, proving a high correlation. Thus, SPRINT can automatically route input data to an optimal $k$ using a simple thresholding method over data smoothness.
>
> **Q4: Inconsistency citation format.**
>
> **A4:** We sincerely apologize for this oversight. We will update the citation entries for all *arXiv* preprints to ensure formatting consistency.
>
> **References:**
>
> [1] Wu, Haixu, et al. "Autoformer: Decomposition transformers with auto-correlation for long-term series forecasting." Advances in neural information processing systems. 2021.
>
> [2] Zeng, Ailing, et al. "Are transformers effective for time series forecasting?." Proceedings of the AAAI conference on artificial intelligence. Vol. 37. No. 9. 2023.

---

> > ### Author Rebuttal · Reviewer_kNnK · 2026-04-02
> >
> > The authors have addressed my comments.

---

> > > ### Author Response · Authors · 2026-04-02
> > >
> > > We are glad that the provided explanations and additional experiments addressed your concerns. Thank you again for your constructive review, which has enhanced the completeness of our work.

---

### Official Review · Reviewer_NYQE · 2026-03-11

**Soundness:** 3
**Presentation:** 3
**Significance:** 3
**Originality:** 3
**Overall Recommendation:** 4
**Confidence:** 3

**Summary:**

This paper proposes SPRINT, a training-free and plug-and-play inference-time framework for long-horizon forecasting with time series foundation models. The main idea is to improve efficiency and accuracy by forecasting in a lower-resolution space rather than directly operating on the full high-resolution sequence. To make this possible without losing too much important information, the method decomposes the input series into trend and seasonal components. The trend is downsampled, forecasted by the backbone TSFM, and then interpolated back to the original resolution, while the seasonal component is extrapolated by replicating representative historical patterns. The method is designed as a black-box wrapper around existing TSFMs and does not require retraining or finetuning. Experiments across several backbones and benchmarks show that SPRINT can substantially reduce inference cost while often improving forecasting accuracy, especially in long-context and long-horizon settings.This paper proposes SPRINT, a training-free and plug-and-play inference-time framework for long-horizon forecasting with time series foundation models. The main idea is to improve efficiency and accuracy by forecasting in a lower-resolution space rather than directly operating on the full high-resolution sequence. To make this possible without losing too much important information, the method decomposes the input series into trend and seasonal components. The trend is downsampled, forecasted by the backbone TSFM, and then interpolated back to the original resolution, while the seasonal component is extrapolated by replicating representative historical patterns. The method is designed as a black-box wrapper around existing TSFMs and does not require retraining or finetuning. Experiments across several backbones and benchmarks show that SPRINT can substantially reduce inference cost while often improving forecasting accuracy, especially in long-context and long-horizon settings.

**Compliance With Llm Reviewing Policy:**

Affirmed.

**Final Justification:**

I keep score at weak accept. The authors have fully addressed my concerns.

**Key Questions For Authors:**

The method depends on decomposition into trend and seasonal components, with the seasonal part forecasted by historical pattern replication. Could the authors clarify more explicitly on what kinds of datasets or failure modes this assumption breaks down, especially when seasonality is weak or unstable?

How sensitive is SPRINT to the quality of decomposition and to the choice of seasonal period? A clearer discussion of robustness here would help assess how reliably the method can be deployed across diverse real-world settings.

The paper shows strong empirical gains across several backbones. In the authors’ view, are these gains mainly due to giving the TSFM an easier low-resolution trend forecasting task, or mainly due to removing the burden of modeling high-frequency seasonal details? A clearer attribution would strengthen the paper’s causal story.

Since SPRINT is positioned as a black-box inference-time wrapper, could the authors discuss whether there are classes of TSFMs or forecasting setups where they would not expect it to work well? This would help better calibrate the scope of the contribution.

The paper is very strong from an efficiency and deployment perspective. Could the authors clarify whether they primarily see the contribution as a practical systems method for TSFMs, or as a more general modeling insight for long-horizon forecasting? This distinction would help interpret the paper’s significance.

**Limitations:**

Yes

**Strengths And Weaknesses:**

Strengths

The paper addresses a real and important practical problem. Long-horizon forecasting with large TSFMs is expensive in time and memory, and the trade-off between longer context and higher inference cost is highly relevant in practice.

The method is simple, coherent, and well motivated. The decomposition into trend and seasonal components is intuitive, and the decision to treat these two parts differently is sensible. The paper does not just downsample blindly, but uses a targeted design in which smooth low-frequency structure is forecasted at low resolution while periodic high-frequency structure is handled through pattern replication.

A major strength is its practical usability. SPRINT is training-free, model-agnostic, and can be wrapped around multiple existing TSFMs without changing their architecture. This gives it clear deployment value and makes the contribution broader than a method tied to one specific backbone.

The empirical gains are meaningful. The paper reports improvements not only in forecasting accuracy, but also very large reductions in memory usage, latency, and compute. This is especially compelling because many papers improve one of these dimensions while sacrificing the others.

The paper also includes useful analysis beyond headline results. It provides evidence for why the approach works, including separate examination of trend interpolation error and seasonal pattern replication. This makes the method feel more principled and less like a pure heuristic trick.

Weaknesses

The main limitation is that the method relies quite heavily on the assumption that the time series can be usefully decomposed into a smooth trend and a repeatable seasonal component. This is reasonable for many datasets, but it may be less effective when seasonality is weak, unstable, drifting, or highly irregular.

Although the method is effective, the core novelty is somewhat limited from a modeling perspective. It is better understood as an inference-time systems enhancement or wrapper around existing TSFMs than as a fundamentally new forecasting model or learning principle.

The theoretical part is helpful for intuition, but it is not especially strong as a rigorous theoretical contribution. The assumptions used to justify interpolation and pattern replication are somewhat idealized, so the paper is more convincing empirically than theoretically.

Because the seasonal component is handled by pattern replication rather than learned forecasting, the method may be less flexible in cases where future high-frequency behavior is not well represented by recent historical cycles. This may limit generalization in more nonstationary or event-driven settings.

---

> ### Author Rebuttal · Authors · 2026-03-30
>
> **Thank you for your kind and careful review!**
>
> **Q1: Failure cases & weak seasonality.**
>
> **A1:** SPRINT's pattern replication assumes high-frequency behavior is quasi-periodic. **In fact, existing literature [1,2] highlights that seasonality is indeed one of the most important factors for LTSF.**  Without seasonality, LTSF becomes very challenging. For instance, DLinear shows that on non-periodic data, even the most advanced models cannot outperform simply copying the most recent data point [2].
>
> Thus, SPRINT may struggle on non-periodic, non-stationary or event-driven data. Despite this, we extend the adaptive strategy of SPRINT in Appendix F to a **data-driven, automated $k$-selection** mechanism. It quantifies "smoothness" to automatically route smoother data to larger downsampling interval $k$, while highly volatile data triggers $k=1$ (bypassing SPRINT). **Please see A3 of our response to Reviewer kNnK for the detailed experimental validation.**
>
> **Q2: Sensitivity to decomposition / $P$.**
>
> **A2:**
> 1. **Moving Average:** SPRINT is robust to the simplicity of moving average (MA). In our ablation study, swapping MA for STL decomposition actually degrades performance and drastically increases runtime. MA functions as a low-pass filter, intentionally leaving non-stationary residuals in trend for the TSFM to resolve.
> 2. **Period ($P$) Selection:** Following established practices in the LTSF literature like SparseTSF [1], $P$ is set via domain knowledge (e.g., 24 for hourly data with daily period).
> 3. **Non-stationarity:** To handle unstable or drifting seasonality, SPRINT does not blindly copy the last cycle. Instead, it computes an weighted average of recent cycles. As shown in our ablation study, the "Season Last" variant (repeating only the recent cycle) yields worse performance. The weighted average effectively integrates historical context, enabling SPRINT to smoothly adapt to pattern shifts and resist local noise.
>
>
> **Q3: Attribution of empirical gains.**
>
> **A3:** The empirical gains are a synergistic combination of **BOTH**:
> 1. **Easier Low-Resolution Task:** In our ablation study, the "w/o Trend Downsampling" variant increases forecasting errors. Thus, downsampling directly benefits the TSFM by condensing long-range dependencies into a shorter context window, simplifying the forecasting task.
> 2. **Removing High-Frequency Burden:** We design a new variant, "Season TSFM", by decomposing the series and force the base TSFM to predict the high-frequency seasonal component, while processing the trend same as SPRINT. As the following table shows, "Season TSFM" produces higher errors than SPRINT across different model architectures. This confirms that removing the burden of modeling high-frequency periodic details is beneficial, preventing TSFMs from overfitting to local noise and computationally wasting capacity.
>
> |Model|Variant|ETTh2|ETTm2|Weather|Service|
> |:-|:-|:-|:-|:-|:-|
> |Moirai|Base|0.372|0.286|0.254|0.304|
> ||+SPRINT|**0.354**|**0.264**|**0.238**|**0.161**|
> ||Season TSFM|0.356|0.267|0.247|0.424|
> |TimesFM|Base|0.402|0.332|/|0.954|
> ||+SPRINT|0.371|**0.278**|/|**0.157**|
> ||Season TSFM|**0.363**|0.290|/|0.588|
> |Timer|Base|0.356|0.279|0.243|0.745|
> ||+SPRINT|**0.343**|**0.254**|**0.227**|**0.494**|
> ||Season TSFM|0.345|0.260|0.231|0.644|
> |VisionTS|Base|0.397|0.320|0.290|**0.426**|
> ||+SPRINT|**0.367**|**0.284**|**0.266**|0.491|
> ||Season TSFM|0.399|0.315|0.292|0.962|
>
>
> **Q4: Limitations on specific TSFMs or setups.**
>
> **A4:**
> 1. **TSFMs:** SPRINT does not yield efficiency improvements for **VLM-based models** like VisionTS, because they treat time series of various lengths as fixed-resolution images. Besides, SPRINT is designed to see a longer history, thus it struggles to improve accuracy for models that strictly require **fixed look-back and prediction lengths**, such as TTM.
> 2. **Setups:** SPRINT offers minimal value in very **short-term forecasting** setups where complexity is not yet a bottleneck.
>
>
> **Q5: Practical systems method vs. general modeling insight.**
>
> **A5:** We primarily position SPRINT as a **practical, training-free, inference-time systems** method designed to overcome the scalability bottlenecks of TSFMs. However, its success exposes a valuable **general modeling insight**: applying downsampling and interpolation exclusively to the low-frequency trend to drastically reduce complexity. SPRINT proves that explicitly injecting the structural prior at the system level is an effective and efficient principle for scalable, ultra-long-term forecasting.
>
> **References:**
>
> [1] Lin, Shengsheng, et al. "SparseTSF: Modeling Long-term Time Series Forecasting with 1k Parameters." In International Conference on Machine Learning, 2024.
>
> [2] Zeng, Ailing, et al. "Are transformers effective for time series forecasting?." Proceedings of the AAAI conference on artificial intelligence. Vol. 37. No. 9. 2023.

---

> > ### Author Rebuttal · Reviewer_NYQE · 2026-04-02
> >
> > clarification is fine.

---

> > > ### Author Response · Authors · 2026-04-02
> > >
> > > We are glad that the provided explanations addressed the concerns. Thank you for your careful review and positive feedback.

---

### Official Review · Reviewer_LYkG · 2026-03-11

**Soundness:** 3
**Presentation:** 3
**Significance:** 3
**Originality:** 3
**Overall Recommendation:** 4
**Confidence:** 4

**Summary:**

This paper proposes SPRINT, a training-free, plug-and-play inference framework, to  alleviate the computational overhead and modeling bottleneck of Time Series
Foundation Models (TSFMs) in long-range forecasting. This method decouples time series data: it downsamples low-frequency trend terms to expand the model's
historical observation window at low cost and uses spline interpolation to restore the  prediction resolution; for high-frequency seasonal terms, it extracts and replicates
historical periodic patterns based on exponential weighting, thus avoiding
information loss caused by downsampling. Empirical results show that this
framework can significantly reduce the memory usage and inference latency of
several existing models while improving their prediction errors.

**Compliance With Llm Reviewing Policy:**

Affirmed.

**Final Justification:**

The rebuttal has addressed my concerns, and I am happy to upgrade.

**Key Questions For Authors:**

1. Advanced TSFMs (e.g., Chronos-2 [1], Moirai2 [2], TabPFN [3]) natively support probabilistic forecasting and exogenous variables. Can the proposed framework seamlessly adapt to these advanced tasks while maintaining its performance gains??
2. Would forcing decomposition on data lacking distinct trend or seasonality lead to performance degradation? Furthermore, in scenarios with negligible trends, the current framework might effectively bypass the TSFM entirely.
3. Certain models (e.g., LightGTS [4]) are specifically designed to capture periodic components to improve forecasting accuracy. Is it theoretically and empirically justified to restrict such models to predict only the low-frequency trend? As by doing this, it may limit the effectiveness of the models.

[1] Chronos-2: From Univariate to Universal Forecasting.

[2] Moirai 2.0: When less is more for time series forecasting.

[3] From Tables to Time: Extending TabPFN-v2 to Time Series Forecasting.

[4] LightGTS: A Lightweight General Time Series Forecasting Model.

**Limitations:**

Yes.

**Strengths And Weaknesses:**

Strengths:
1. The motivation is clear: it effectively addresses the significant issues of Time Series Fundamental Models (TSFM) in long-range forecasting, namely excessive computational overhead and limited input
sequence length.
2. It has theoretical support: it provides theoretical analysis and proof regarding the effectiveness of the trend term downsampling mechanism and the upper bound of the cumulative error for seasonal term
pattern replication.

Weaknesses:
1. Lack of support for probabilistic forecasting: The current framework is restricted to point forecasting, leaving its compatibility with probabilistic or interval forecasting unverified.
2. Insufficient reliability justification: The absence of error bars and statistical significance tests raises concerns about the stability and robustness of the reported performance gains.
3. Limited evaluation metrics: The experiments rely solely on MSE, completely omitting other standard and critical metrics such as MAE.
4. Limitations of the decomposition mechanism: Forcing this decomposition approach on time series that lack clear seasonal or trend components might actually degrade overall performance.

---

> ### Author Rebuttal · Authors · 2026-03-31
>
> **Thank you for your careful review!**
>
> **W1 & Q1: Compatibility with probabilistic forecasting and exogenous variables.**
>
> **A1:** **SPRINT seamlessly supports probabilistic forecasting and exogenous variables.** Specifically, if the base TSFM outputs a predictive distribution (e.g., sampled trajectories), our resolution interpolation workflow is simply applied independently to each predicted trajectory to restore the high-resolution probabilistic forecast. Exogenous variables are similarly downsampled during the look-back window.
>
> Due to time limit, we conduct experiments on Chronos and TimesFM using the widely-used **CRPS** (continuous ranked probability score). As shown below, SPRINT-enhanced TSFMs consistently outperform their bases, confirming its robust predictive superiority. A slash (/) denotes datasets included in the model’s pretraining.
>
> |Model|Variant|ETTh2|ETTm2|Weather|Service|
> |:-|:-|:-|:-|:-|:-|
> |Chronos|Base|0.369|0.320|0.257|0.584|
> ||+SPRINT|**0.351**|**0.282**|**0.236**|**0.265**|
> |TimesFM|Base|0.295|0.246|/|0.457|
> ||+SPRINT|**0.272**|**0.227**|/|**0.277**|
>
> **W2: Insufficient reliability justification.**
>
> **A2:** In fact, **error bars and statistical significance tests are not applicable to our zero-shot setting**. SPRINT is a _training-free, zero-shot inference wrapper_. When applied to deterministic TSFMs for zero-shot point forecasting, the outputs are deterministic. Thus, running experiments across multiple random seeds yields **identical** results. The reported performance gains are stable for the given model checkpoints.
>
> **W3: Limited evaluation metrics.**
>
> **A3:** For visual clarity in our primary tables, we focus on MSE in the main text. In fact, SPRINT shows a **consistent performance gain** in MAE. We present a selection of these results here, while the full results will be included in the Appendix. A slash (/) denotes datasets included in the model’s pretraining.
>
> |Model|Variant|ETTh2|ETTm2|Weather|Service|
> |:-|:-|:-|:-|:-|:-|
> |Chronos|Base|0.403|0.353|0.292|0.619|
> ||+SPRINT|**0.397**|**0.322**|**0.277**|**0.222**|
> |TimesFM|Base|0.409|0.342|/|0.612|
> ||+SPRINT|**0.405**|**0.335**|/|**0.228**|
> |TimeMoE|Base|0.386|0.339|0.365|0.483|
> ||+SPRINT|**0.383**|**0.316**|**0.268**|**0.225**|
> |VisionTS|Base|0.425|0.371|0.330|**0.395**|
> ||+SPRINT|**0.403**|**0.342**|**0.312**|0.536|
>
> **W4 & Q2: Data lacking distinct trend/seasonality.**
>
> **A4:** SPRINT adapts to diverse data types and does not simply "bypass" the TSFM when trends are negligible:
> - **Lacking a distinct trend:** The moving average still acts as a low-pass filter. Forecasting this smoothed, non-periodic component with the TSFM prevents the model from overfitting to high-frequency noise, yielding significantly better performance than feeding it raw, noisy sequences.
> - **Lacking distinct seasonality:** Existing literature shows that LTSF inherently relies heavily on seasonality [1, 2]. Advanced models often cannot outperform a naive baseline (repeat last point) on non-periodic data [2]. Despite this, for non-periodic data, we extend the adaptive strategy in Appendix F to an **automated, data-driven $k$-selection** mechanism. By quantifying "smoothness", it routes smoother data to larger downsampling interval $k$, while highly volatile, non-periodic data triggers $k=1$ (bypassing SPRINT). **Please see A3 to Reviewer kNnK for the detailed experimental validation.**
>
> **Q3: Application for models specifically designed for periodicity.**
>
> **A5:** Even for models optimized for periodicity (e.g., LightGTS), SPRINT proves essential for *high-frequency-sampled data*.
>
> |Model|Variant|ETTh2|ETTm2|Weather|Service|
> |:-|:-|:-|:-|:-|:-|
> |LightGTS|Base|**0.360**|0.339|0.307|0.780|
> ||+SPRINT|0.406|**0.289**|**0.256**|**0.473**|
>
> Under the experimental settings in the main paper, results indicate that while SPRINT offers no improvement on hourly data (ETTh2), it delivers accuracy gains as the sampling frequency increases (ETTm2, Weather, Service). Because higher sampling rates require much longer sequences to capture the same **physical context**, SPRINT’s downsampling mechanism effectively expands the **receptive field** of base TSFMs, thereby enhancing their performance.
>
> **References:**
>
> [1] Lin, Shengsheng, et al. "SparseTSF: Modeling Long-term Time Series Forecasting with 1k Parameters." In International Conference on Machine Learning, 2024.
>
> [2] Zeng, Ailing, et al. "Are transformers effective for time series forecasting?." Proceedings of the AAAI conference on artificial intelligence. Vol. 37. No. 9. 2023.

---

> > ### Author Rebuttal · Reviewer_LYkG · 2026-04-02
> >
> > The authors have addressed my comments. I am happy to upgrade.

---

> > > ### Author Response · Authors · 2026-04-02
> > >
> > > It is encouraging to know that our supplementary analyses successfully resolved your concerns. We sincerely appreciate your constructive advice and support.

---

### Official Review · Reviewer_fmvx · 2026-03-11

**Soundness:** 2
**Presentation:** 3
**Significance:** 3
**Originality:** 2
**Overall Recommendation:** 4
**Confidence:** 4

**Summary:**

This paper proposes SPRINT (Seasonal Pattern Replication and Trend Resolution INTerpolation), a training-free inference-time plugin for improving long-term and ultra-long-term time series forecasting with existing time-series foundation models (TSFMs). The key idea is to decompose an input series into trend and seasonality, then treat them differently: the smooth trend component is downsampled, forecast by the base TSFM at low resolution, and interpolated back to the original resolution, while the seasonal component is forecast by extracting a representative periodic pattern from recent cycles and replicating it over the prediction horizon. In this way, SPRINT aims to let TSFMs access effectively longer contexts at lower computational cost without retraining the underlying model. Experiments across multiple TSFMs and benchmark datasets show consistent gains in both forecasting accuracy and efficiency, with particularly strong improvements on data with clear periodic structure, while the paper also discusses limitations on non-periodic or rough series and suggests an adaptive variant that uses only the resolution-interpolation component in such cases.

**Compliance With Llm Reviewing Policy:**

Affirmed.

**Final Justification:**

The authors have sufficiently addressed my concerns, so I maintain my original positive score.

**Key Questions For Authors:**

Please refer to the weaknesses.

**Limitations:**

yes

**Strengths And Weaknesses:**

## Strengths

- Leveraging long historical context and improving long-horizon forecasting are central challenges in time series forecasting, especially for modern TSFMs whose computational cost grows quickly with input and prediction length. The paper tackles this problem in a simple and effective way by combining interpolation for the trend component and replication for the seasonal component, allowing the model to access longer effective context while reducing inference cost.
- The trend-resolution interpolation strategy is grounded in an intuitive theoretical argument: after decomposition, the trend is treated as a smooth low-frequency signal that can be downsampled and later reconstructed with limited loss.
- The motivation, method, ablations, and efficiency analysis are presented in a coherent manner, making the paper easy to follow. The empirical section is also strong overall: the method shows consistent improvements on well-known quasi-periodic benchmark datasets across multiple TSFMs.

## Weaknesses
- The seasonal pattern replication module relies on a fairly strong quasi-periodicity assumption. While the authors acknowledge this limitation and propose an alternative strategy in Appendix F, the method still appears most applicable to datasets with clear and stable periodic structure. This makes the scope of the approach somewhat narrow, especially for many real-world forecasting settings where seasonality is weak, shifting, or non-stationary.
- Since the method explicitly decomposes the input into trend and seasonality and handles the two parts differently, it would strengthen the empirical case to compare against a baseline that also decomposes the series into trend and seasonal components, forecasts each component separately using the backbone TSFM, and then recombines them. Such a comparison would help clarify whether the gains come specifically from the proposed interpolation/replication design, rather than from decomposition alone.
- The sentence “As evidenced by frequency-domain compression and filtering strategies, complex signals can often be accurately reconstructed from a reduced set of critical components.” seems to make a fairly broad technical claim and would benefit from an explicit citation.
- The experimental coverage could be broader. It would be helpful to include results on widely used forecasting benchmarks such as Electricity, Traffic, and Illness, which are common reference datasets in the time series literature. In addition, I would be interested in seeing how the method behaves on financial/stock data, where the assumptions behind SPRINT are less likely to hold.
- The paper would benefit from more quantitative evaluation on datasets that violate its assumptions, as well as a clearer adaptive criterion. While Appendix F discusses non-periodic settings through a smoothness-based adaptive strategy, this remains somewhat heuristic. More systematic experiments on datasets that do not satisfy the assumptions of SPRINT would make the limitations clearer.

---

> ### Author Rebuttal · Authors · 2026-03-30
>
> **Thank you for your kind and careful review!**
>
> **Q1 & Q5 & Q6: Limitation for data with weak seasonality & adaptive strategy for non-periodic data & performance on Financial/Stock Data.**
>
> **A1:** For data without seasonality, LTSF is very challenging. Numerous literature [1, 2] highlights that **seasonality is indeed one of the most important factors for LTSF.**  DLinear shows that on non-periodic data, even the most advanced models cannot outperform simply copying last point [2].
>
> Since SPRINT may struggle on non-periodic data, we extend the adaptive strategy to a **data-driven, automated $k$-selection** mechanism. It quantifies "smoothness" to automatically route smoother data to larger downsampling interval $k$, while highly volatile data triggers $k=1$ (bypassing SPRINT). The smoothness, $s$, is quantified as the product of the Turning Points Ratio and Normalized Volatility, where a lower value indicates a smoother data. We conduct an analysis on the correlation between $k_{opt}$ and $s$, using 3 financial/stock datasets which are day-sampled and non-periodic. To get different $s$, we apply moving average on raw datasets and get smoothed variants (e.g., `S4` denotes an MA window of 4). Results show that smoother data consistently correlates with a larger $k_{opt}$. The Pearson correlation coefficient between $\log_2(k_{opt})$ and $s$ is **-0.859**, proving a high correlation.
>
> |Dataset|Variant|Smoothness(1e-3)$\downarrow$|$k_{opt}$|MSE|
> |:-|:-|:-|:-|:-|
> |Exchange|Raw|10.5|1|0.0395|
> ||S2|5.60|2|0.0379|
> ||S4|2.85|2|0.0386|
> ||S8|1.38|4|0.0396|
> ||S16|0.720|8|0.0396|
> |NASDAQ|Raw|12.3|1|1.040|
> ||S2|7.12|1|1.080|
> ||S4|4.36|4|1.096|
> ||S8|2.62|4|0.955|
> ||S16|1.50|4|0.829|
> |NYSE|Raw|6.15|4|0.637|
> ||S2|3.73|4|0.638|
> ||S4|2.33|4|0.639|
> ||S8|1.40|8|0.644|
> ||S16|0.853|8|0.599|
>
> On **raw** financial/stock data, while SPRINT may underperform the base TSFM ($k=1$), it shows superior accuracy when applied to **moving-averaged** data, which better captures **long-term financial trends**.
>
> We will include these analysis with experimental results in the appendix.
>
> **Q2: Comparison with variant: forecasting both components with the TSFM.**
>
> **A2:** We decompose the series and force the base TSFM to predict the seasonal component (denoted as "Season TSFM"), while the trend is processed same as SPRINT. The results, compared to both base TSFMs and SPRINT, are shown below, with a slash (/) indicating datasets used in the model’s pretraining.
>
> |Model|Variant|ETTh2|ETTm2|Weather|Service|
> |:-|:-|:-|:-|:-|:-|
> |Moirai|Base|0.372|0.286|0.254|0.304|
> ||+SPRINT|**0.354**|**0.264**|**0.238**|**0.161**|
> ||Season TSFM|0.356|0.267|0.247|0.424|
> |TimesFM|Base|0.402|0.332|/|0.954|
> ||+SPRINT|0.371|**0.278**|/|**0.157**|
> ||Season TSFM|**0.363**|0.290|/|0.588|
> |Timer|Base|0.356|0.279|0.243|0.745|
> ||+SPRINT|**0.343**|**0.254**|**0.227**|**0.494**|
> ||Season TSFM|0.345|0.260|0.231|0.644|
> |VisionTS|Base|0.397|0.320|0.290|**0.426**|
> ||+SPRINT|**0.367**|**0.284**|**0.266**|0.491|
> ||Season TSFM|0.399|0.315|0.292|0.962|
>
> "Season TSFM" yields higher errors compared to SPRINT. However, it still outperforms the base TSFM. This confirms that SPRINT's effectiveness stems from **BOTH the decomposition strategy and the pattern replication mechanism**.
>
> **Q3: Citation for the signal reconstruction claim.**
>
> **A3:** Thanks! We will append citations on a standard signal processing textbook [3] alongside Shannon's sampling theorem [4] to explicitly support our claim.
>
> **Q4: Experiments on more datasets.**
>
> **A4:** In the main paper, we deliberately exclude Electricity and Traffic to maintain a zero-shot setting. Numerous TSFMs have trained on these datasets (Chronos, TimesFM, TimeMoE, ToTo for Electricity, and Moirai, TimesFM, TimeMoE for Traffic).
>
> Despite this, here we present results on these datasets. Due to space and time limit, we conduct experiments on selected TSFMs with different architectures. The prediction length of Illness is set as $L\in\{24,36,48,60\}$ following existing literature. The look-back length is set as $H=30P$ consistent as our main setting, except $H=3P$ for Illness for its short overall length. A slash (/) denotes datasets included in the model’s pretraining.
>
> |Model|Variant|Illness|Electricity|Traffic|
> |:-|:-|:-|:-|:-|
> |Moirai|Base|2.325|**0.178**|/|
> ||+SPRINT|**2.099**|0.195|/|
> |TimesFM|Base|2.189|/|/|
> ||+SPRINT|**1.968**|/|/|
> |VisionTS|Base|2.707|1.021|1.609|
> ||+SPRINT|**2.011**|**0.217**|**0.470**|
>
>
> **References:**
>
> [1] Lin, Shengsheng, et al. "SparseTSF: Modeling Long-term Time Series Forecasting with 1k Parameters." In International Conference on Machine Learning, 2024.
>
> [2] Zeng, Ailing, et al. "Are transformers effective for time series forecasting?." Proceedings of the AAAI conference on artificial intelligence. 2023.
>
> [3] Oppenheim, Alan V. Discrete-time signal processing. Pearson Education India, 1999.
>
> [4] Shannon, Claude E. "Communication in the presence of noise." Proceedings of the IRE 37.1 (2006): 10-21.

---

> > ### Author Rebuttal · Reviewer_fmvx · 2026-04-04
> >
> > Thank you for the detailed response. My concerns have been sufficiently addressed, and therefore I will maintain my original positive score.

---

> > > ### Author Response · Authors · 2026-04-05
> > >
> > > We are glad that our response sufficiently addressed your concerns. Thank you again for your careful review and positive feedback.

---

### Decision · Program_Chairs · 2026-04-30

**Decision:**

Accept (regular)

**Comment:**

This paper proposes SPRINT, a training-free and plug-and-play inference-time framework for long-horizon forecasting with time series foundation models. The main idea is to improve efficiency and accuracy by forecasting in a lower-resolution space rather than directly operating on the full high-resolution sequence. To make this possible without losing too much important information, the method decomposes the input series into trend and seasonal components. The trend is downsampled, forecasted by the backbone TSFM, and then interpolated back to the original resolution.
Authors successfully resolved the most of concerns raised by four reviews during the rebuttal.

Although the paper is good to present as it is, the paper does not run against complex real-world time series datasets. Thus, it is recommend when the trend prediction (with interpolation) is not accurate enough (failure cases). Additionally, it would be worthwhile to include qualitative examples when the proposed method outperform existing models.